# Regulatory consequences of neuronal ELAV-like protein binding to coding and non-coding RNAs in human brain

Claudia Scheckel[1,2†], Elodie Drapeau[3,4†], Maria A Frias[1,2], Christopher Y Park[1,2,5], John Fak[1,2], Ilana Zucker-Scharff[1,2], Yan Kou[3,6], Vahram Haroutunian[4,7,8], Avi Ma'ayan[6], Joseph D Buxbaum[3,4,9,10,11*‡], Robert B Darnell[1,2,5*‡]

[1]Laboratory of Molecular Neuro-Oncology, The Rockefeller University, New York, United States; [2]Howard Hughes Medical Institute, The Rockefeller University, New York, United States; [3]Seaver Autism Center for Research and Treatment, New York, United States; [4]Department of Psychiatry, Icahn School of Medicine at Mount Sinai, New York, United States; [5]New York Genome Center, New York, United States; [6]Department of Pharmacology and Systems Therapeutics, BD2K-LINCS Data Integration and Coordination Center, Mount Sinai Knowledge Management Center for Illuminating the Druggable Genome, Icahn School of Medicine at Mount Sinai, New York, United States; [7]Department of Neuroscience, Icahn School of Medicine at Mount Sinai, New York, United States; [8]James J. Peters VA Medical Center, New York, United States; [9]Friedman Brain Institute, Icahn School of Medicine at Mount Sinai, New York, United States; [10]Mindich Child Health Institute, Icahn School of Medicine at Mount Sinai, New York, United States; [11]Department of Genetics and Genomic Sciences, Icahn School of Medicine at Mount Sinai, New York, United States

*For correspondence: joseph.
buxbaum@mssm.edu (JDB);
darnelr@rockefeller.edu (RBD)

†These authors contributed
equally to this work
‡These authors also contributed
equally to this work

Competing interests: The
authors declare that no
competing interests exist.

Reviewing editor: Douglas L
Black, University of California,
Los Angeles, United States

**Abstract** Neuronal ELAV-like (nELAVL) RNA binding proteins have been linked to numerous neurological disorders. We performed crosslinking-immunoprecipitation and RNAseq on human brain, and identified nELAVL binding sites on 8681 transcripts. Using knockout mice and RNAi in human neuroblastoma cells, we showed that nELAVL intronic and 3' UTR binding regulates human RNA splicing and abundance. We validated hundreds of nELAVL targets among which were important neuronal and disease-associated transcripts, including Alzheimer's disease (AD) transcripts. We therefore investigated RNA regulation in AD brain, and observed differential splicing of 150 transcripts, which in some cases correlated with differential nELAVL binding. Unexpectedly, the most significant change of nELAVL binding was evident on non-coding Y RNAs. nELAVL/Y RNA complexes were specifically remodeled in AD and after acute UV stress in neuroblastoma cells. We propose that the increased nELAVL/Y RNA association during stress may lead to nELAVL sequestration, redistribution of nELAVL target binding, and altered neuronal RNA splicing.

## Introduction

RNA binding proteins (RBPs) associate with RNAs throughout their life cycle, regulating all aspects of RNA metabolism and function. More than 800 RBPs have been described in human cells (*Castello et al., 2012*). The unique structure and function of neurons, and the need to rapidly adapt RNA regulation in the brain both within and at sites distant from the nucleus, are consistent with

**eLife digest** When a gene is active, its DNA is copied into a molecule of RNA. This molecule then undergoes a process called splicing which removes certain segments, and the resulting 'messenger RNA' molecule is then translated into protein. Many messenger RNAs go through alternative splicing, whereby different segments can be included or excluded from the final molecule. This allows more than one type of protein to be produced from a single gene.

Specialized RNA binding proteins associate with messenger RNAs and regulate not only their splicing, but also their abundance and location within the cell. These activities are crucially important in the brain where forming memories and learning new skills requires thousands of proteins to be made rapidly. Many members of a family of RNA binding proteins called ELAV-like proteins are unique to neurons. These proteins have also been associated with conditions such as Alzheimer's disease, but it was not known which messenger RNAs were the targets of these proteins in the human brain.

Scheckel, Drapeau et al. have now addressed this question and used a method termed 'CLIP' to identify thousands of messenger RNAs that directly bind to neuronal ELAV-like proteins in the human brain. Many of these messenger RNAs coded for proteins that are important for the health of neurons, and neuronal ELAV-like proteins were shown to regulate both the alternative splicing and the abundance of these messenger RNAs.

The regulation of RNA molecules in post-mortem brain samples of people with or without Alzheimer's disease was then compared. Scheckel, Drapeau et al. unexpectedly observed that, in the Alzheimer's disease patients, the neuronal ELAV-like proteins were very often associated with a class of RNA molecules known as Y RNAs. These RNA molecules do not code for proteins, and are therefore classified as non-coding RNA. Moreover, massive shifts in the binding of ELAV-like proteins onto Y RNAs were observed in neurons grown in the laboratory that had been briefly stressed by exposure to ultraviolet radiation.

Scheckel, Drapeau et al. suggest that the strong tendency of neuronal ELAV-like proteins to bind to Y RNAs in conditions of short- or long-term stress, including Alzheimer's disease, might prevent these proteins from associating with their normal messenger RNA targets. This was supported by finding that some messenger RNAs targeted by neuronal ELAV-like proteins showed altered regulation after stress. Such changes to the normal regulation of these messenger RNAs could have a large impact on the proteins that are produced from them.

Together, these findings link Y RNAs to both neuronal stress and Alzheimer's disease, and suggest a new way that a cell can alter which messenger RNAs are expressed in response to changes in its environment. The next step is to explore what causes the shift in neuronal ELAV-like protein binding from messenger RNAs to Y RNAs and how it might contribute to disease.

specialized roles for RBPs in the brain. Indeed, mammalian neurons have developed their own system of RNA regulation (*Darnell, 2013*), and RBP:mRNA interactions are thought to regulate local protein translation at synapses, perhaps underlying learning and long-term memory (*McKee et al., 2005*).

Numerous RBPs have been linked to human neurological disorders (reviewed in *Richter and Klann (2009)*). For example, *FUS, TDP-43* and *ATXN2* mutations have been found in familial amyotrophic lateral sclerosis patients (*Elden et al., 2010*; *Vance et al., 2009*; *Sreedharan et al., 2008*), *TDP-43* has additionally been associated with frontotemporal lobar degeneration, Alzheimer's Disease (AD) and Parkinson's Disease (PD) (*Baloh, 2011*), *STEX* has been linked to amyotrophic lateral sclerosis 4 (*Chen et al., 2004*), and spinal muscular atrophy can be caused by mutations in *SMN* (*Clermont et al., 1995*).

The neuronal ELAV-like (ELAVL) and NOVA RBPs are targeted by the immune system in paraneoplastic neurodegenerative disorders (*Buckanovich et al., 1996*; *Szabo et al., 1991*). Mammalian ELAVL proteins include the ubiquitously expressed paralog ELAVL1 (also termed HUA or HUR) and the three neuron-specific paralogs, ELAVL2, 3 and 4 (also termed HUB, C, and D, and collectively referred to as nELAVL; *Ince-Dunn et al., 2012*). nELAVL proteins are expressed exclusively in

neurons in mice (*Okano and Darnell, 1997*), and they are important for neuronal differentiation and neurite outgrowth in cultured neurons (*Akamatsu et al., 1999*; *Kasashima et al., 1999*; *Mobarak et al., 2000*; *Anderson et al., 2000*; *Antic et al., 1999*; *Aranda-Abreu et al., 1999*). Redundancy between the three nELAVL isoforms complicates in vivo studies of their individual functions. Nevertheless, even haploinsuffiency of *Elavl3* is sufficient to trigger cortical hypersynchronization, and *Elavl3* and *Elavl4* null mice display defects in motor function and neuronal maturation, respectively (*Akamatsu et al., 2005*; *Ince-Dunn et al., 2012*).

ELAVL proteins have been shown to regulate several aspects of RNA metabolism. In vitro and in tissue culture cells, nELAVL proteins have been implicated in the regulation of stabilization and/or translation of specific mRNAs, as well as in the regulation of splicing and polyadenylation of select transcripts [reviewed in *Pascale et al. (2004)*]. A more comprehensive approach was taken by immunoprecipitating an overexpressed isoform of ELAVL4 in mice, although such RNA immunoprecipitation experiments cannot distinguish between direct and indirect targets (*Bolognani et al., 2010*). Recently, direct binding of nELAVL to target RNAs in mouse brain was demonstrated by high-throughput sequencing of RNA isolated by crosslinking immunoprecipitation (HITS-CLIP; *Ince-Dunn et al., 2012*); these data, coupled with transcriptome profiling of *Elavl3/4* KO mice, demonstrated that nELAVL directly regulates neuronal mRNA abundance and alternative splicing by binding to U-rich elements with interspersed purine residues in 3'UTRs and introns in mouse brain (*Ince-Dunn et al., 2012*).

While genome-wide approaches have been applied to studying nELAVL proteins in mice, the targets of nELAVL in the human brain remain largely unknown. This is of particular importance, as nELAVL proteins have been implicated in neurological disorders such as AD (*Amadio et al., 2009*; *Kang et al., 2014*) and PD (*DeStefano et al., 2008*; *Noureddine et al., 2005*). Hence, to advance our understanding of the function of nELAVL in humans and its link to human disease, we set out to investigate nELAVL:RNA interactions in the human brain.

To globally identify transcripts directly bound by nELAVL in human neurons, we generated a genome wide RNA binding map of nELAVL in human brain using CLIP. CLIP allows the identification of functional RNA-protein interactions in vivo by using UV-irradiation of intact tissues to covalently crosslink and then purify RNA-protein complexes present in vivo (*Licatalosi and Darnell, 2010*; *Ule et al., 2003*). This method has been adopted for a variety of RBPs (*Darnell, 2010*; *2013*; *Moore et al., 2014*). Here, we systemically identified tens of thousands of reproducible nELAVL binding sites in human brain and showed that nELAVL binds transcripts that are important for neurological function and that have been linked to neurological diseases such as AD. We validated the functional consequences of nELAVL binding in mice and cultured human neuroblastoma cells and showed that the loss of nELAVL affected mRNA abundance and alternative splicing of hundreds of transcripts. We further investigated RNA regulation in AD brains, and found that numerous transcripts were differentially spliced in AD, which correlated with differential nELAVL binding in some cases. Remarkably, we observed the most significant increase in nELAVL binding in AD on a class of non-coding RNAs, Y RNAs. We recapitulated these findings in human neuroblastoma cells, showing that nELAVL binding is linked to Y ribonucleoprotein (RNP) remodeling acutely during UV-induced stress, and chronically in AD.

## Results

### Identification of nELAVL targets in human brain

To gain insight into nELAVL-mediated RNA regulation in human brain we performed CLIP on post-mortem brain samples of eight human subjects (*Supplementary file 1A*). Tissue samples were derived from BA9, which is part of the dorsolateral prefrontal cortex (*Figure 1A*), a brain area that is damaged in later stages of AD and that is important for executive functions such as working memory, cognitive flexibility, planning, inhibition, and abstract reasoning (*O'Reilly, 2010*). Antibodies that specifically recognize individual ELAVL paralogs and that can be used for CLIP are currently not available. We therefore purified nELAVL-RNP complexes with antiserum reactive to all three nELAVL proteins (*Figure 1B*). $^{32}$P-labeled nELAVL-RNP complexes were not recovered with control serum or in the absence of UV-irradiation (*Figure 1B*).

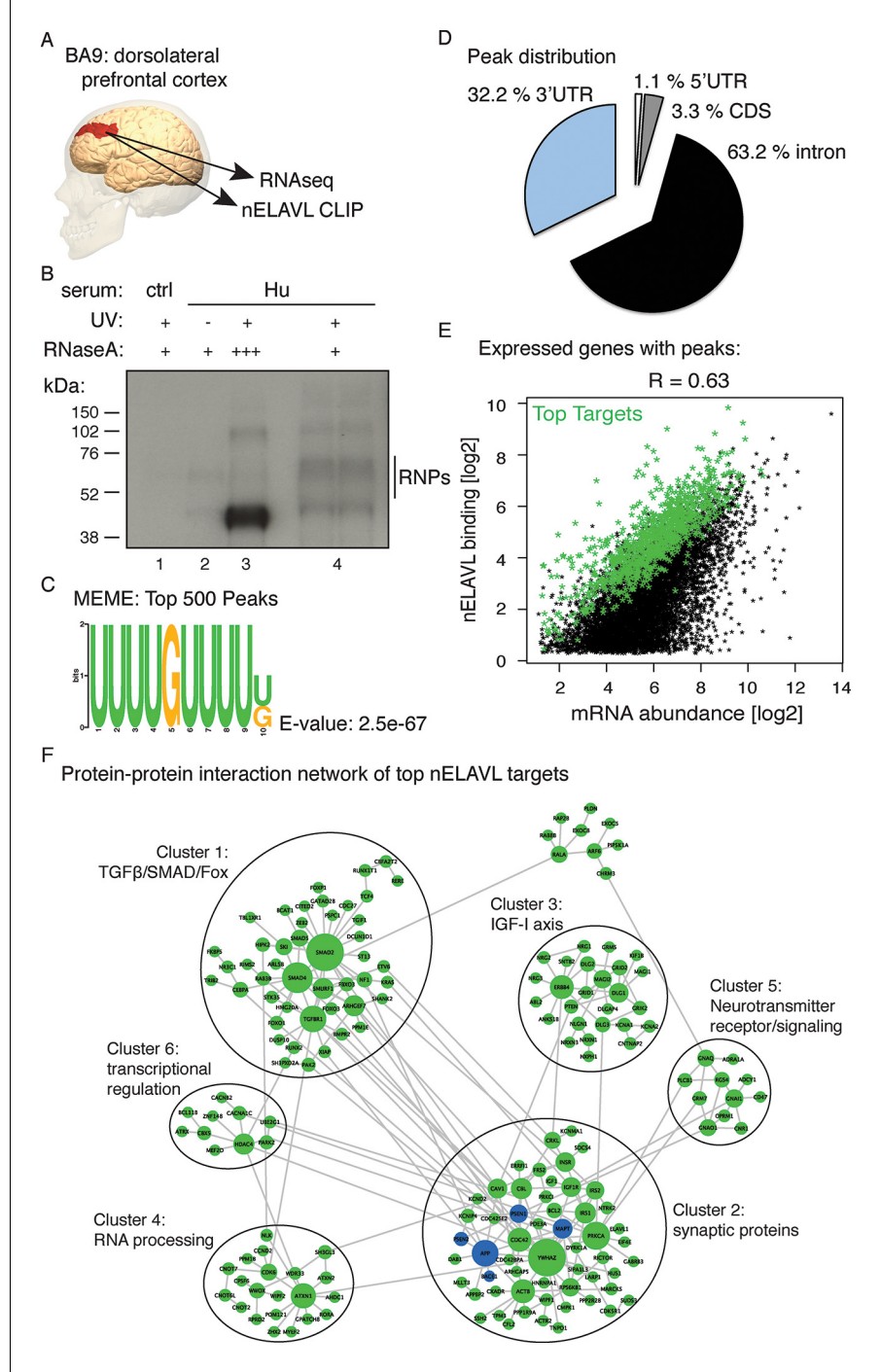

**Figure 1.** Identification of nELAVL targets in human brain. (**A**) Illustration depicting the brain area analyzed by CLIP and RNAseq. The image was generated using BodyParts3D/Anatomography service by DBCLS, Japan. (**B**) SDS-PAGE separation of radiolabeled nELAVL-RNP complexes. nELAVL-RNP complexes from 40 mg of human brain were specifically immunoprecipitated with Hu-antiserum, compared to control serum (compare lane #4 to #1), which is dependent on UV irradiation (compare lane #4 to #2). Wide-range nELAVL-RNP complexes collapse to a single band in the presence of high RNAse concentration (lane #3). RNAse dilutions: + 19.23 Units/µl; +++ 3846 Units/µl. As in studies of mouse nELAVL (*Ince-Dunn et al., 2012*), higher molecular weight bands were present in nELAVL CLIP autoradiograms, which correspond at least in part to nELAVL multimers. (**C**) Shown is the most enriched motif in the top 500 nELAVL peaks, determined with MEME-ChiP. (**D**) Pie chart of the genomic peak distribution of 75,592 nELAVL peaks (p < 0.01; present in at least 5 individuals). (**E**) nELAVL binding correlates with mRNA abundance. nELAVL binding (CLIP tags within binding sites per transcript) was compared to mRNA

*Figure 1 continued on next page*

*Figure 1 continued*

abundance (RNAseq tags per transcript). Only expressed genes with peaks are shown and the correlation coefficient is indicated. The top 1000 targets were identified as genes with highest normalized nELAVL binding (binding sites were normalized for mRNA abundance and summarized per gene). (F) Subnetwork of direct protein-protein interactions of top nELAVL targets. The 1000 top nELAVL target genes and six additional genes highly associated with AD (APP, BACE1, MAPT, PICALM, PSEN1 and PSEN2) were clustered using the organic layout algorithm in yEd. Genes with no direct interactions with other target genes were excluded, leaving 172 nodes from the top nELAVL target list (green) and 5 AD associated genes (blue) in this subnetwork. The size of the nodes is proportional to the connectivity degree. Six clusters (gray circles) containing at least 10 nodes were identified, and subjected to enrichment analysis (see *Supplementary file 1F*).

The following figure supplements are available for figure 1:

**Figure supplement 1.** Cross-correlation plot comparing nELAVL peak binding between eight individuals (n = 75,592).

**Figure supplement 2.** Shown is the most enriched motif in the top 500 nELAVL peaks, determined with HOMER.

**Figure supplement 3.** Pie chart of the genomic peak distribution of 75,592 nELAVL peaks (p < 0.01; present in at least 5 individuals), normalized for region length.

**Figure supplement 4.** nELAVL peaks within 3'UTRs are higher than intronic binding sites.

**Figure supplement 5.** Cross-correlation plot comparing the mRNA abundance of all transcripts between eight individuals (n = 19,185).

**Figure supplement 6.** Correlations between mRNA abundance and nELAVL binding.

---

nELAVL-crosslinked RNA tags were sequenced and mapped to the hg18 build of the human genome. We searched for and identified nELAVL RNA binding sites (peaks) that were significant (p<0.01) and that were present in at least five out of 8 individuals (n = 75,592). nELAVL binding at these sites correlated between individuals (*Figure 1—figure supplement 1*), and these peaks were further investigated. We determined the nELAVL binding motif by analyzing the top 500 nELAVL peaks (+/- 25nt) using MEME ChIP (*Machanick and Bailey, 2011*) and HOMER (*Heinz et al., 2010*). This revealed that nELAVL binds polyU RNA stretches in human brain, particularly when interrupted by a G (*Figure 1C* and *Figure 1—figure supplement 2*). This is in excellent agreement with the nELAVL binding motif identified in mouse brain using CLIP and in vitro binding assays (*Ince-Dunn et al., 2012*). Because nELAVL has been shown to regulate alternative splicing and mRNA abundance in mouse brain by binding to introns and 3'UTRs, respectively, we analyzed the genomic distribution of the peaks defined here. As previously reported for mouse nElavl (*Ince-Dunn et al., 2012*), nELAVL binding sites are found in 3'UTRs and introns (*Figure 1D*), with far higher per-nucleotide density in 3'UTR regions (*Figure 1—figure supplement 3*). Consistently we observed that nELAVL peaks in 3'UTRs were higher than intronic peaks (*Figure 1—figure supplement 4*). These results suggest that nELAVL could regulate splicing and mRNA abundance in the human brain.

To relate nELAVL binding to mRNA abundance, we performed RNAseq on the same brain samples used for CLIP analysis (*Figure 1—figure supplement 5* and *Supplementary file 1A/B*). 74,423 nELAVL peaks mapped to 8681 expressed genes (*Supplementary file 1C*), referred to as nELAVL targets (*Supplementary file 1D*) hereafter, which are shown in *Figure 1E*. We observed that nELAVL binding correlated with mRNA abundance (*Figure 1E*). This was not unexpected, as nELAVL 3' UTR binding has previously been shown to increase mRNA abundance, due to its role in mRNA stabilization. The correlation between nELAVL binding and mRNA abundance might therefore not only reflect the dependence of nELAVL binding on mRNA abundance, but also a role of nELAVL in mRNA stabilization. Consistently, we observed that intronic nELAVL binding correlated less with mRNA abundance than 3'UTR binding (*Figure 1—figure supplement 6*).

To identify genes most likely to be impacted by nELAVL, we defined the top 1000 nELAVL targets (colored in green in *Figure 1E*; *Supplementary file 1E*). Top targets were identified based on

normalized nELAVL binding (binding sites were normalized for mRNA abundance and summarized per gene). Thirty-seven percent of nELAVL peaks (n = 27,581) mapped to these top targets. We constructed a subnetwork that connects top nELAVL target gene products based on a literature-based network of protein-protein interactions (PPI) created from multiple online databases (*Chen et al., 2012*). Six clusters were identified within the resulting network (*Figure 1F*) and gene set enrichment analyses were performed for top nELAVL targets found in the different clusters with Enrichr (*Chen et al., 2013*). Each cluster was examined for enrichment of Biological Processes (BP), Molecular Functions (MF), Cellular Components, and Mammalian Phenotypes (MP) terms (*Supplementary file 1F*). Enriched terms included RNA processing and transcription regulation, signal transduction, synaptic transmission, synaptic proteins, and abnormal neuron morphology and physiology. Three of the six clusters were particularly important for neuronal function. Cluster 1 is especially enriched in members of the TGFbeta/SMAD signaling pathways and the FOX protein family. Cluster 2 contains many actors of the IGF-I axis, which is important for neuronal development including neurogenesis, myelination, synaptogenesis, dendritic branching and neuroprotection after neuronal damage. Finally, cluster 3 is almost exclusively formed of synaptic proteins including many postsynaptic scaffolding proteins, members of the neuroligin/neurexin families and glutamatergic receptors or voltage-gated channels. Taken together, these data demonstrate that nELAVL associates with transcripts encoding proteins involved in key aspects of neuronal physiology.

## nELAVL-mediated regulation is conserved

To investigate if nELAVL-mediated RNA regulation is conserved, we compared our dataset with previously published nELAVL targets in mice (*Ince-Dunn et al., 2012*; *Bolognani et al., 2010*). At the transcript level, we found that more than 90% of mouse nELAVL targets were among human nELAVL targets (*Figure 2A*). However, only 20% of human targets were bound by nELAVL in mouse brain, which is at least partly due to the 10-fold increased depth of the human dataset (more than 10 million human CLIP tags compare to less than a million mouse CLIP tags). Yet these differences could also reflect an increased functional complexity of nELAVL regulation in the human brain, and/or the fact that mouse targets were identified at different developmental stages.

We additionally investigated the overlap of nELAVL binding at individual binding sites. Surprisingly, only a small percentage of binding sites overlapped between mouse and human; 3% of human binding sites showed nELAVL binding in mouse, and 17% of mouse binding sites were bound by nELAVL in human brain (*Figure 2B*). The vast majority of these overlapping binding sites were in 3'UTRs (88%). These results indicate that many nELAVL targets are shared between mouse and human and that nELAVL binding at the transcript level is conserved, whereas individual binding sites have diverged drastically, especially within introns. These results reflect analogous observations of evolutionary conservation of transcriptional regulation at the gene rather than the positional level (*Stergachis et al., 2014*).

We further observed that nELAVL binding on entire transcripts correlated between mouse and human (*Figure 2—figure supplement 1*), which prompted us to overlay our human CLIP dataset with previously published transcriptome profiling of *Elavl3/4* double KO mice (*Ince-Dunn et al., 2012*). 119 transcripts showed significant changes in their steady-state level in *Elavl3/4* KO mice, 91 of which were expressed in human brain. 37 of these 91 transcripts were nELAVL 3'UTR targets in human brain (*Supplementary file 2A*), and the majority of them decreased in the absence of ELAVL3/4 (n = 26), including transcripts important for neuronal transport and excitation such as RAB6B, HCN3, and KCNMB2 (*Figure 2C/D*). This indicates that nELAVL 3'UTR binding is likely to be important for increasing the abundance of these transcripts in human brain, and likely has conserved functions across species.

*Elavl3/4* KO mice also reported splicing defects and 59 alternative exons showed a significant change in their inclusion rate (delta inclusion rate, ΔI) between wildtype and *Elavl3/4* KO mice (*Ince-Dunn et al., 2012*). 54 of the misregulated exons were conserved in the human genome, and 25 of them were adjacent to intronic nELAVL binding sites in human brain (*Supplementary file 2B*). We observed both increased and decreased inclusion of alternative exons – independently of the position of the peak relative to the exon, which has previously been observed for nELAVL mediated splicing regulation (*Ince-Dunn et al., 2012*). Three alternative exons are shown in *Figure 2E,F*, and whereas nELAVL seems to prevent splicing of *DST* by binding upstream and downstream of an alternative exon, nELAVL might promote the inclusion of alternative exons of *NRXN1* and *CELF2* by

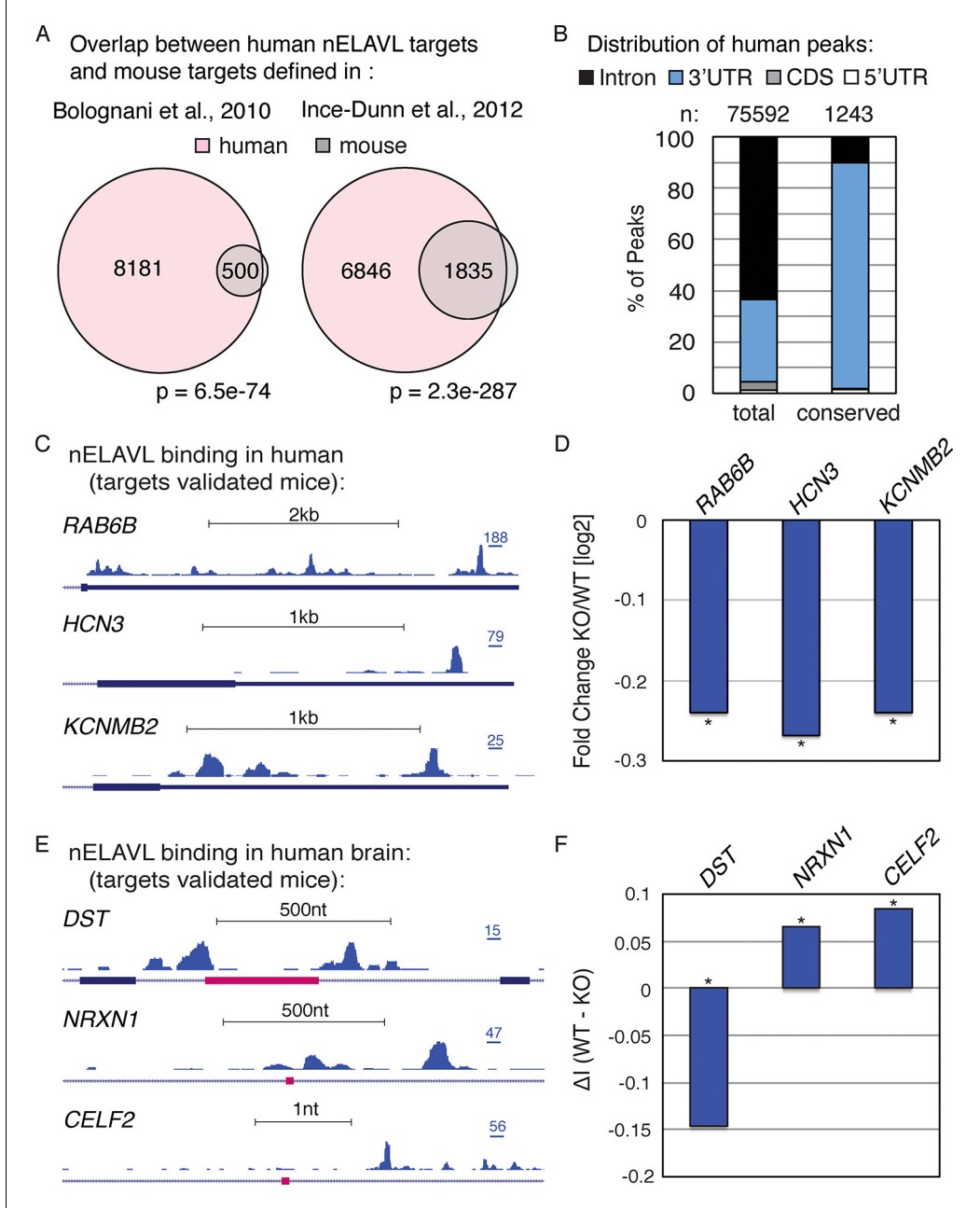

**Figure 2.** nELAVL mediated regulation is conserved in mouse and human. (**A**) Overlap of nELAVL targets in human and mouse. Human nELAVL targets (n = 8681) were intersected with mouse targets identified by RIP (*Bolognani et al., 2010*) or HITS-CLIP (*Ince-Dunn et al., 2012*). 538 genes were identified as nELAVL targets by RIP and were expressed in human brain. 1978 expressed genes had HITS-CLIP nELAVL clusters that were present in at least 3 samples (biological complexity (BC) ≥ 3). Both overlaps (n = 500 and n = 1835) were highly significant (p = 6.5e$^{-74}$ and p = 2.3e$^{-287}$; hypergeometric test), compared to expressed transcripts (n = 14,737). (**B**) Only few nELAVL binding sites are conserved between mice and human, which are predominantly present within 3'UTRs. The genomic distribution of all human nELAVL binding sites (total) and nELAVL binding sites conserved in mouse is shown. The number of nELAVL binding sites (n) within each category is indicated. (**C**) UCSC Genome Browser images illustrating the 3'UTRs of *RAB6B, HCN3*, and *KCNMB2* and their normalized nELAVL binding profile in human brain. The maximum PeakHeight is indicated by numbers in the right corner. (**D**) The mRNA levels of transcripts with nELAVL 3'UTR binding decrease in *Elavl3/4* knockout (KO) mice. Shown are the mRNA expression fold changes (knockout/wildtype) of *RAB6B, HCN3*, and *KCNMB2*. *p< 0.01 (two-tailed t test; *Ince-Dunn et al., 2012*). (**E**) UCSC Genome Browser images showing pink cassette exons in the *DST, NRXN1*, and *CELF2* genes and

*Figure 2 continued on next page*

*Figure 2 continued*

their normalized nELAVL binding profiles in human brain. The maximum PeakHeight is indicated by numbers in the right corner. (**F**) nELAVL binding adjacent to a cassette exon in the *DST* gene prevents exon inclusion. Downstream nELAVL binding promotes the inclusion of cassette exons in the *NRXN1* and *CELF2* genes. The change in alternative exon inclusion (delta inclusion (ΔI): wildtype - *Elavl3/4* KO) is shown. * significantly changing (analyzed by Aspire2; *Ince-Dunn et al., 2012*).

The following figure supplement is available for figure 2:

**Figure supplement 1.** Comparison of nELAVL binding (CLIP tags within binding sites per transcript) between mice and human.

binding to downstream sequences. Given that nELAVL regulates the splicing of these 25 exons in mice and that we observe intronic nELAVL binding sites in human brain adjacent to these exons, we propose that nELAVL regulates the inclusion of these exons in human brain. Collectively, these analyses show that many confirmed functional nELAVL interactions in mouse brain show evidence for nELAVL binding in human brain.

## nELAVL proteins regulate mRNA abundance of human brain targets

To further validate potential nELAVL targets, we analyzed the effect of nELAVL depletion on mRNA abundance and splicing in human neuroblastoma IMR-32 cells. We subjected IMR-32 cells to mock or *ELAVL2/3/4* triple RNAi, achieving 70% knockdown of all three neuronal ELAVL proteins (*Figure 3—figure supplement 1*). These cells were then analyzed by RNAseq (*Figure 3A*, *Supplementary file 1A/2C*). The steady-state level of 784 transcripts was significantly changed in *nELAVL* RNAi treated cells (*Figure 3A*), with ~45% showing a decrease in mRNA abundance. Among those genes were all of the neuronal ELAVL paralogs (*ELAVL2/3/4*), while the ubiquitously expressed paralog ELAVL1 was not affected by *nELAVL* RNAi depletion.

We then compared this RNAseq dataset with the nELAVL CLIP analysis in human brain. 96% of the IMR-32 expressed transcripts were expressed in human brain (12,242 out of 12,743), and were used for subsequent analyses. Since nELAVL binding to 3'UTRs can mediate mRNA stabilization, we investigated the change in mRNA abundance of 3'UTR targets upon *nELAVL* RNAi, specifically examining 3'UTR targets that did not display any intronic binding. These transcripts were less abundant in *nELAVL* RNAi conditions (*Figure 3B*, left panel). In contrast, the mRNA abundance of intron targets (intron binding but no 3'UTR binding) slightly increased upon *nELAVL* RNAi (*Figure 3B*, right panel). This suggests that specifically nELAVL binding to 3'UTRs increases mRNA abundance. We further observed that nELAVL depletion affected top nELAVL 3'UTR targets as well as nELAVL 3'UTR targets in general (*Figure 3C*).

Out of 784 genes that changed significantly upon *nELAVL* RNAi, 743 genes were expressed in human brain, 327 of which decreased while 416 increased. We investigated which of these transcripts were direct targets of nELAVL based on nELAVL 3'UTR binding (*Supplementary file 2D*). Significantly changing transcripts that are top nELAVL 3'UTR targets are boxed in blue in *Figure 3D*. We observed that 68% of downregulated transcripts were 3'UTR targets (n = 226; p = $9.6e^{-13}$; hypergeometric test), and that 16% of downregulated transcripts were even among top 3'UTR targets (n = 51; p = $1.3e^{-6}$; hypergeometric test). In contrast, only 7% of upregulated transcripts were among top 3'UTR targets (p = 0.76; hypergeometric test), further supporting a role of nELAVL 3'UTR binding in positively regulating mRNA abundance. Several 3'UTR targets showed an mRNA abundance change in both mouse and IMR-32 datasets (n = 8), and in all but one case this change correlated positively between the datasets, providing support for the accuracy of target validation applied here. Because the abundance of multiple disease associated genes, including *APPBP2, ATXN3*, and *SHANK2* (*Figure 3E,F*), is regulated by nELAVL, we propose that nELAVL mediated regulation of mRNA abundance plays an important role in the human brain.

## nELAVL regulates splicing of human brain targets

To validate a role of nELAVL in the splicing regulation of human brain targets, we analyzed the inclusion rate of cassette exons in mock and *nELAVL* RNAi treated IMR-32 cells. We compared the

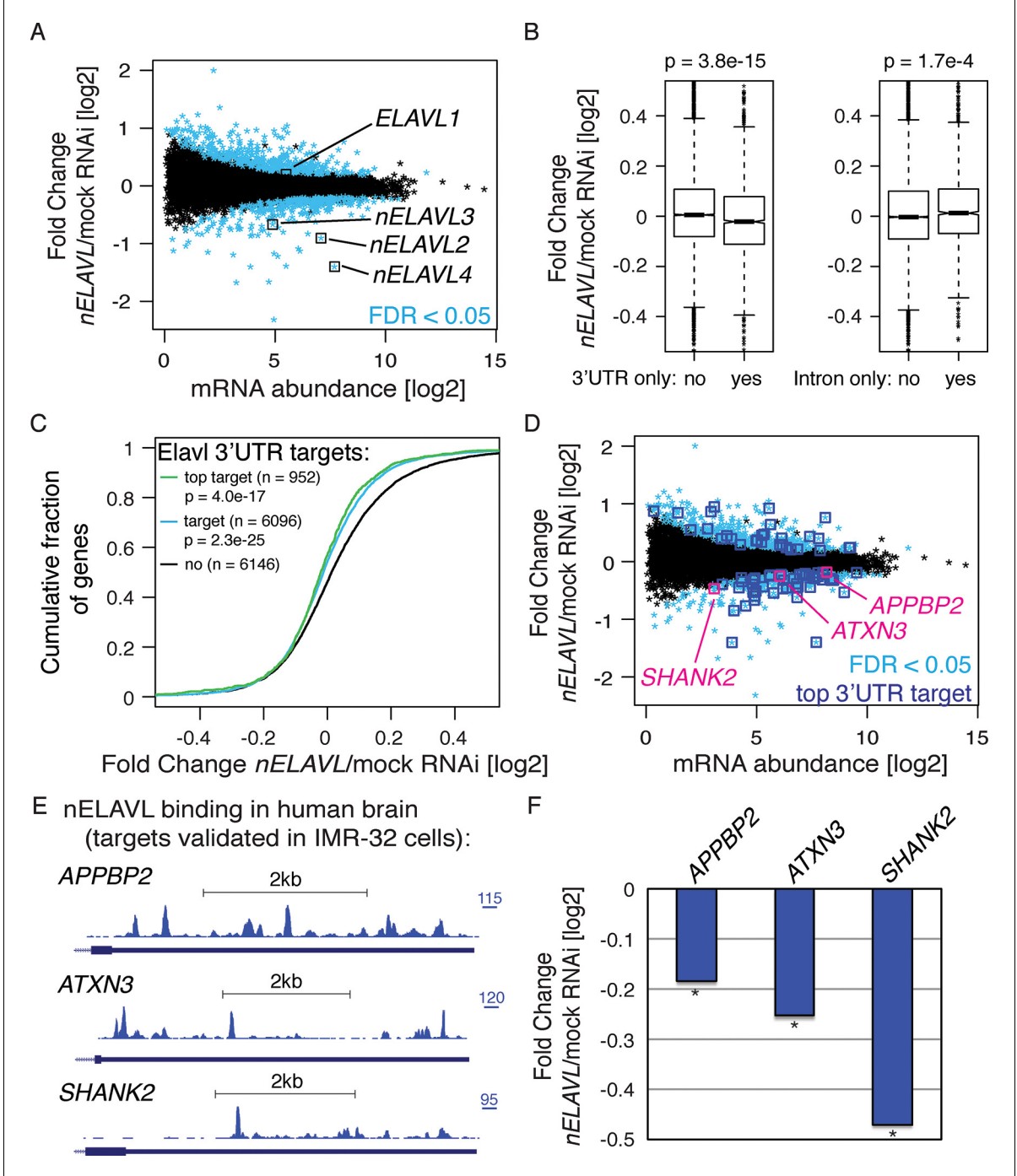

**Figure 3.** nELAVL proteins regulate mRNA abundance of human brain targets. (**A**) nELAVL depletion causes mRNA level changes in IMR-32 neuroblastoma cells. The mRNA abundance change was plotted against average mRNA abundance. Significantly changing transcripts (FDR < 0.05; n = 784) are colored in blue. Shown are only expressed genes (n = 12,743), and *ELAVL1/2/3/4* transcripts are indicated. (**B**) nELAVL with exclusively 3'UTR binding decrease upon *nELAVL* RNAi depletion. Box plots represent the distribution of mRNA level differences between mock and *nELAVL* RNAi. We compared genes with exclusively 3'UTR (n = 2346) or intronic (n = 1693) binding that were expressed in IMR-32 cells. nELAVL binding was defined as CLIP tags within binding sites per transcript. Transcripts with exclusively 3'UTR binding were less abundant upon *nELAVL* RNAi compared to remaining transcripts (p = $3.8e^{-15}$; two-tailed t-test). In contrast, mRNA levels of transcripts with exclusively intron binding were even slightly increased compared to remaining transcripts (p = $1.7e^{-4}$; two-tailed t-test). (**C**) Transcripts with nELAVL 3'UTR binding decrease upon *nELAVL* RNAi. Cumulative fraction curves for genes with no 3'UTR nELAVL binding in human brain, 3'UTR binding, and top 3'UTR targets. Top targets were identified as 1000 genes with highest normalized nELAVL 3'UTR binding (binding sites were normalized for mRNA abundance before summarized per gene). 952 of the top 1000 targets were expressed in IMR-32 cells. A curve displacement to the left indicates a downregulation of mRNA abundance upon *nELAVL* RNAi. p values

*Figure 3 continued on next page*

Figure 3 continued

were calculated with a one-sided KS test, comparing (top) targets to non-targets. (D) Many transcripts that are decreasing upon nELAVL depletion are top nELAVL 3'UTR targets. The mRNA abundance change (nELAVL/mock RNAi) of transcripts expressed in IMR-32 cells and in human brain (n = 12,242) was plotted against average mRNA abundance. Significantly changing transcripts (FDR<0.05; n = 743) are colored in blue and additionally boxed if they are top nELAVL 3'UTR targets. Transcripts shown in E/F are indicated. (E) UCSC Genome Browser images illustrating the 3'UTRs of APPBP2, ATXN3, and SHANK2 and their normalized nELAVL binding profile in human brain. The maximum PeakHeight is indicated by numbers in the right corner. (F) The mRNA abundance of top nELAVL 3'UTR targets decreases upon nELAVL RNAi. Shown are the mRNA level changes (nELAVL/mock RNAi) of APPBP2, ATXN3, and SHANK2. * FDR<0.05 (derived from edgeR).
The following figure supplement is available for figure 3:

Figure supplement 1. Western blot and its quantification showing protein levels of nELAVL and the housekeeping genes HSP90 and Histone H3 in mock and nELAVL RNAi-treated IMR-32 cells.

change in exon inclusion of 7903 expressed cassette exons (*Supplementary file 2E*) and observed that 473 cassette exons were differentially spliced upon nELAVL depletion (FDR<0.05 and ∆I > 0.1; *Supplementary file 2F*). Many differentially spliced exons were adjacent (+/- 2.5 kb) to at least one intronic nELAVL binding site in human brain (n = 155; p = $1.3e^{-7}$; hypergeometric test; *Figure 4A*, *Supplementary file 2F*), indicating that these exons might be directly regulated by nELAVL. For example, downstream binding in *BIN1* and *PICALM* was associated with lower exon inclusion upon nELAVL depletion, and binding in *APP* was associated with higher inclusion of both upstream and downstream exons upon nELAVL depletion (*Figure 4B/C*). Overall, three exons that were differentially spliced upon *nELAVL RNAi* depletion also changed in *Elavl3/4* KO mice, and the splicing changes in both datasets changed in the same direction. We generated a map from intronic nELAVL binding sites that flanked the 155 nELAVL regulated exons as previously described (*Licatalosi et al., 2008*), revealing that upstream nELAVL binding can promote both exon inclusion and skipping (*Figure 4D*). In conclusion, these data indicate that intronic nELAVL binding regulates alternative splicing of numerous transcripts in human brain, including transcripts associated with central nervous system disorders.

## RNA regulation changes in AD

nELAVL has previously been linked to neurological diseases and we observed that nELAVL regulated the mRNA abundance and splicing of multiple disease-associated genes. We examined nELAVL binding in a set of genes with disease associated 3'UTR single nucleotide polymorphisms (SNPs) (*Bruno et al., 2012*). We found that these genes were enriched among nELAVL 3'UTR targets (n = 200; p = 0.001; hypergeometric test), and that nELAVL binding sites directly overlapped with 45 disease associated SNPs, including SNPs associated with autism, schizophrenia, depression, AD, and PD (*Figure 5—figure supplement 1*, *Supplementary file 3A*).

nELAVL proteins have been implicated in AD (*Amadio et al., 2009*; *Kang et al., 2014*), and among the validated nELAVL regulated RNAs were also several AD-related transcripts, which led us to investigate additional AD-linked genes (hereafter termed AD genes; n = 96; *Supplementary file 3B*). Indeed, we found that the top nELAVL targets were enriched among AD genes (n = 11; p = 0.03; hypergeometric test; contained in *Supplementary file 3B*) as well as among AD risk loci identified in a genome-wide association study (GWAS) in AD (*Naj et al., 2011*) (n = 77; p = $1.7e^{-14}$; hypergeometric test; *Supplementary file 3C*). To investigate if nELAVL mediated regulation of AD related and other transcripts might be affected in AD, we performed nELAVL CLIP and RNAseq on AD subject brains, age-matched to control subjects (*Figure 5—figure supplement 2*, *Supplementary file 1A/B* and *3D*). Importantly, ELAVL3/4 mRNA levels were similar between control and AD samples and ELAVL2 showed only a slight decrease in transcript abundance in AD brains (*Supplementary file 1B*), which allowed us to compare nELAVL binding profiles between control and AD brains. We did not detect many significant changes in nELAVL binding nor mRNA abundance (*Figure 5A/B*, *Supplementary file 1B* and *3D*), probably due to the variation between human samples, the small sample size, and the potential heterogeneity of AD. We did however observe that 150 transcripts were differentially spliced in the 9 AD subjects (FDR<0.05 and ∆I>0.1; *Figure 5C*, *Supplementary file 3E*). Two of these transcripts, *BIN1* and *PTPRD*, have previously

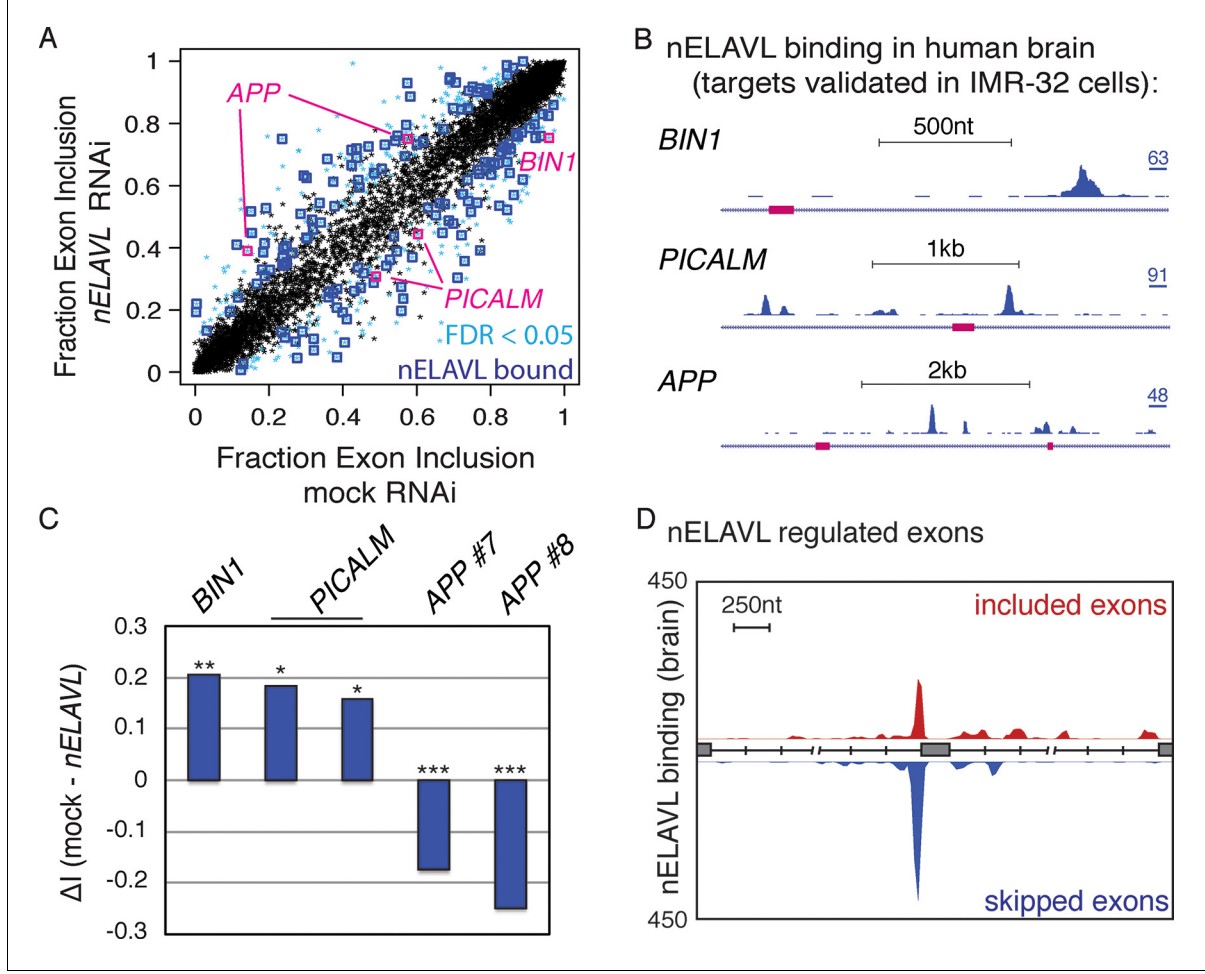

**Figure 4.** nELAVL regulates splicing of human brain targets. (**A**) Analysis of splicing changes upon *nELAVL* RNAi. Shown is the exon inclusion fraction of cassette exons that are expressed in IMR-32 cells and in human brain (n = 7903). Significantly changing exons (FDR<0.05 and ΔI>0.1) are colored in light blue (n = 473), and additionally boxed in dark blue if adjacent (+/- 2.5 kb) to intronic nELAVL binding sites (n = 155). Significantly changing exons shown in (**B/C**) are boxed in pink. The two alternative events within *PICALM* correspond to the same alternative exon with two different 3' splice sites. (**B**) UCSC Genome Browser images depicting cassette exons in pink in the *BIN1*, *PICALM*, and *APP* genes and their normalized nELAVL binding profiles in human brain. The maximum PeakHeight is indicated by numbers in the right corner. (**C**) nELAVL binding downstream of cassette exons in *BIN1* and *PICALM* promotes exon inclusion, whereas intronic nELAVL binding of *APP* prevents exon inclusion downstream and upstream. The change in alternative exon inclusion (ΔI: mock – *nELAVL* RNAi) is shown. *FDR< 0.0005; **FDR< 1e$^{-4}$; ***FDR<1e$^{-16}$ (GLM likelihood ratio test). (**D**) Normalized nELAVL binding map of nELAVL regulated exons. Only exons that changed significantly upon *nELAVL* RNAi (FDR<0.05 and ΔI>0.1) and that are adjacent (+/- 2.5 kb) to intronic nELAVL binding sites (n = 155) were included. Red and blue peaks represent binding associated with nELAVL-dependent exon inclusion and exclusion, respectively.

been linked to AD (*Tan et al., 2013*; *Ghani et al., 2012*), suggesting that the differential splicing of these two transcripts as well as other RNAs might be linked to AD.

As shown above (*Figure 4*), nELAVL depletion in IMR-32 cells was associated with the reduced inclusion of an alternative exon of *BIN1*, suggesting that nELAVL binding promotes the inclusion of this exon. Precisely this exon was differentially spliced in AD subjects, with AD subjects showing a reduced exon inclusion rate compared to control subjects (*Figure 5D*). Along with the differential exon inclusion, we observed that nELAVL peak binding was fourfold decreased in AD subjects (log2 fold change = -2.35; p = 0.16; *Figure 5D*). These results are consistent with nELAVL-mediated dysregulation of this exon in AD subjects, with decreased binding leading to decreased exon inclusion. In conclusion, while we did not detect global nELAVL binding and mRNA abundance changes in AD subjects, we observed that splicing of 150 transcripts was affected, which in some cases might be linked to nELAVL dysregulation.

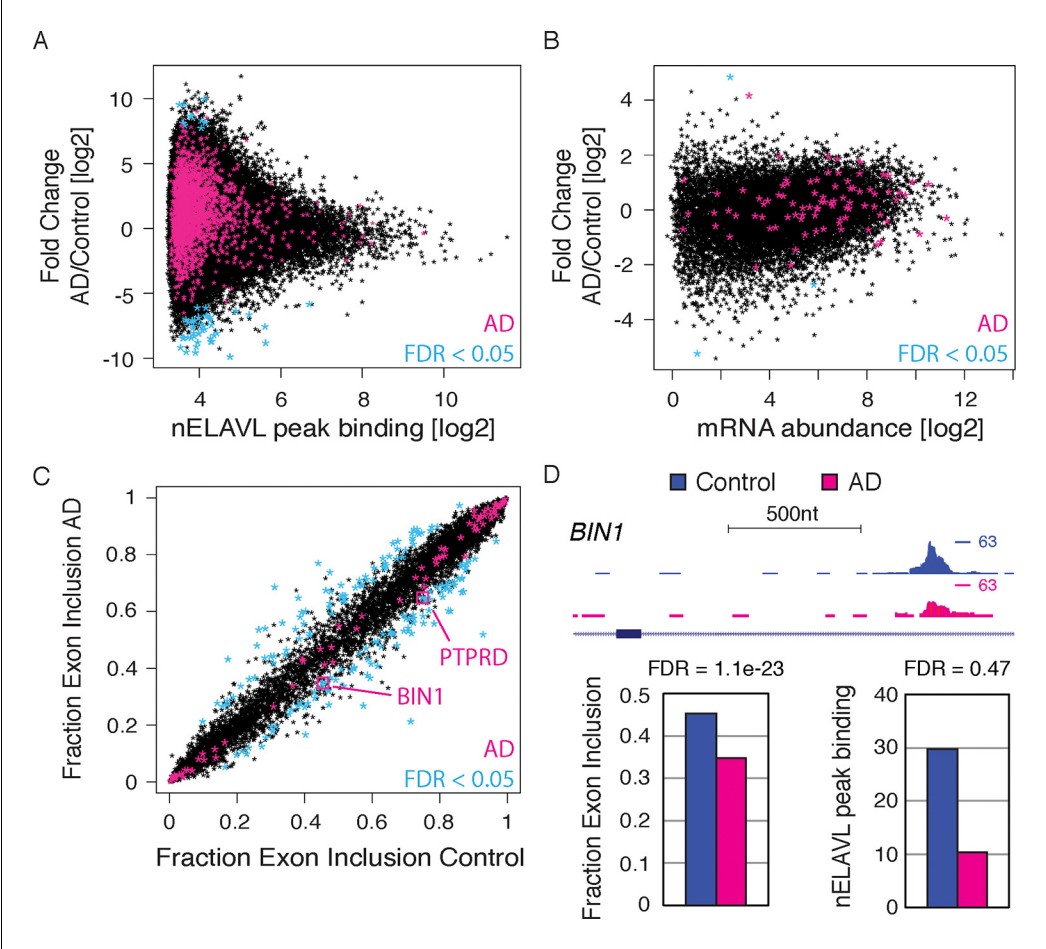

**Figure 5.** RNA regulation changes in AD. (**A**) nELAVL binding changes in AD. The nELAVL peak binding change (AD/Control) was plotted against average nELAVL peak binding. Significantly changing peaks (FDR<0.05; n = 52) are colored in blue, and peaks within AD genes are colored in pink (1811 peaks within 69 genes). Shown are only peaks that are bound in control or AD brain (n = 115,393). (**B**) mRNA abundance changes in AD. The mRNA abundance change (AD/Control) was plotted against average mRNA abundance. Significantly changing transcripts (FDR<0.05; n = 3) are colored in blue, and AD transcripts are colored in pink (n = 89). Shown are only transcripts that are expressed in control or AD brain (n = 14,875). (**C**) Analysis of splicing changes in AD. Shown is the inclusion fraction of expressed cassette exons in control and AD subjects (n = 8163). Exons within AD genes are colored in pink (n = 79). Significantly changing exons (FDR<0.05 and ΔI>0.1) are colored in light blue (n = 170), and additionally boxed in pink if within AD genes (n = 2). (**D**) *BIN1* is alternatively spliced in AD. UCSC Genome Browser image illustrating a cassette exon in the *BIN1* gene and normalized nELAVL binding profiles in control and AD brain. The maximum PeakHeight is indicated by numbers in the right corner. Bar graphs depict the difference in alternative exon inclusion (ΔI: Control – AD) and nELAVL peak binding (AD/Control) in control and AD brain. Corresponding FDR values derived from edgeR are shown. The inclusion of the exon is promoted by nELAVL (see *Figure 4*), and exon inclusion as well as nELAVL peak binding are reduced in AD subjects.

The following figure supplements are available for figure 5:

**Figure supplement 1.** Examples of disease associated SNPs with corresponding nELAVL binding sites.

**Figure supplement 2.** Correlations between control and AD samples.

## Non-coding Y RNAs are bound by nELAVL in AD

The largest fold changes in nELAVL binding in AD (relative to the age-matched control population) occurred on a specific class of non-coding RNAs, Y RNAs (*Wolin et al., 2013*). Y RNAs are 100 nt long structured RNAs usually found in complex with RO60 (also known as TROVE2; *Figure 6A*; modified from *Chen and Wolin, 2004*). RO60 is believed to act as a sensor of RNA quality, targeting defective RNAs for degradation (*Sim and Wolin, 2011*). RO60 was initially identified as an

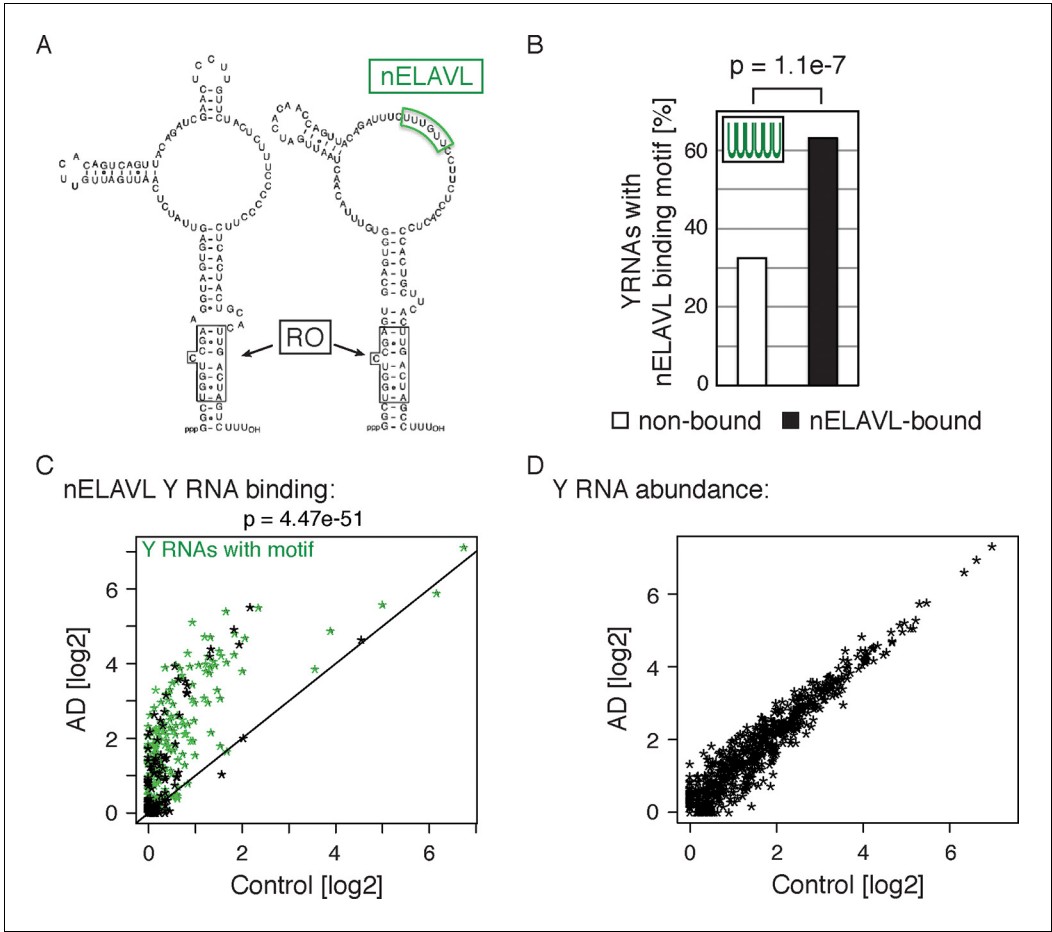

**Figure 6.** Non-coding Y RNAs are bound by nELAVL in AD. (**A**) Secondary structures of Y1 and Y3. Binding sites of nELAVL and Ro are indicated. Modified from (**Chen and Wolin, 2004**). (**B**) The nELAVL binding motif (UUUUUU, allowing a G at any position) is enriched in nELAVL-bound Y RNAs compared to non-bound Y RNAs (p = 1.1e-7; Fisher's exact test). Y RNAs were scanned for (T)$_6$, allowing a G at any position. nELAVL-bound Y RNAs: nELAVL CLIP tags in at least two samples; n = 320. (**C**) nELAVL binding of Y RNAs increases in AD compared to control samples (p = 4.47e-51; paired one-sided Wilcoxon rank sum test). The axes depict nELAVL Y RNA binding (nELAVL CLIP tags per Y RNA) in control and AD subjects. Y RNAs with nELAVL binding motif are colored in green. (**D**) Y RNA levels do not change in AD. Y RNA abundance (RNAseq tags per Y RNA) in AD subjects was plotted against Y RNA abundance in control subjects.

The following figure supplements are available for figure 6:

**Figure supplement 1.** Y RNAs with a motif that are not bound are not expressed.

**Figure supplement 2.** nELAVL:Y RNA binding increases in AD.

autoantigen targeted in systemic lupus (**Lerner et al., 1981**) and some subjects with the paraneo-plastic encephalopathy syndrome harbor both anti-RO and anti-nELAVL (Hu) autoantibodies (**Manley et al., 1994**). Four canonical Y RNAs, Y1/3/4/5, have been characterized in humans, but numerous slightly divergent copies of these Y RNAs, especially Y1 and Y3, are distributed throughout the human genome (**Perreault et al., 2005**).

Surprisingly, we observed nELAVL binding to a total of 320 Y RNAs, although Y RNA copies other than the canonical four Y1/3/4/5 genes had previously been considered to be non-functional and were labeled 'pseudogenes' (**Supplementary file 3F**). We found that 237 of the 320 nELAVL bound Y RNAs were Y3-like RNAs (**Supplementary file 3F**), and that nELAVL bound Y RNAs showed an enrichment of the nELAVL binding motif (202 Y RNAs contained UUUUUU, allowing a G at any one position), which is also present in the canonical hY3 RNA (**Figure 6A/B**). We examined the 118 nELAVL bound Y RNAs that did not fit this consensus in more detail. 91 of these Y RNAs (77%)

contained either a 5mer version of the motif or the motif with an A or C instead of a G, and we found U/G rich stretches in the remaining 27 Y RNAs (*Supplementary file 3F*). In addition, some Y RNAs with a strong binding motif did not show any evidence of nELAVL binding. In general, these Y RNAs showed a lower expression compared to nELAVL bound Y RNA, which may explain the absence of detectable nELAVL binding (*Figure 6—figure supplement 1*).

We next explored nELAVL/Y RNA binding in AD brain. We observed a drastic increase in nELAVL/Y RNA association in AD subjects (*Figure 6C*), while Y RNA levels remained largely unchanged (*Figure 6D*). This suggests that Y RNPs undergo nELAVL-dependent remodeling in AD. Interestingly, we did observe a high variability in nELAVL/Y RNA association between AD samples (*Figure 6—figure supplement 2*), with three of them showing a very strong nELAVL/Y RNA association. Efforts to relate this difference to the expression of stress-related genes, post-mortem interval, age, extent of disease and cause of death were not conclusive, and the cause for the variation in nELAVL binding to Y RNAs among AD subjects remains elusive.

## Y RNPs are remodeled during UV stress

The observation of increased nELAVL/Y RNA association in AD raised the possibility that Y RNP remodeling is associated with neuronal stress. Y RNP remodeling has previously been linked to UV-induced stress (*Sim et al., 2009*), and both bacterial (*Chen et al., 2000*; *Wurtmann and Wolin, 2010*) and mouse cells (*Chen et al., 2003*) show an increased sensitivity to UV stress in the absence of RO60. ELAVL binding can be modulated in response to stress in cultured cells (*Bhattacharyya et al., 2006*), and ELAVL proteins, which shuttle between nucleus and cytoplasm in response to environmental cues, preferentially accumulate in cytoplasmic stress granules upon stress (*Gallouzi et al., 2000*; *Fan and Steitz, 1998b*). We therefore examined the effect of acute UV stress on Y RNP remodeling in IMR-32 cells. IMR-32 cells were exposed to a low dose of UV stress (not sufficient to induce RNA:protein crosslinking) and allowed to recover for 24 h before being analyzed by nELAVL CLIP. We found that nELAVL bound 132 Y RNAs in neuroblastoma cells (*Supplementary file 3F*), that Y RNAs showed an enrichment of the nELAVL binding motif (*Figure 7A*) or at least contained a degenerate version of it (*Supplementary file 3F*), and that non-bound Y RNAs with a motif show a very low expression (*Figure 7—figure supplement 1*). Moreover, nELAVL binding on Y RNAs was dynamic and increased in UV stressed cells compared to non-stressed cells (*Figure 7B* and *Figure 7—figure supplement 2*), while their abundance did not change upon UV irradiation (*Figure 7C*). To assess whether Y RNA levels were affected by nELAVL, we depleted *nELAVL* by RNAi three days prior to the UV exposure, and analyzed Y RNA levels by RNAseq. Y RNA abundance was not affected by nELAVL depletion in UV stressed IMR-32 cells (*Figures 7D*). These results indicate that increased nELAVL binding to Y RNAs is not a function of Y RNA levels, and that nELAVL binding during stress is not required for Y RNA stability.

To determine if UV stress induced localization changes of Y RNP components, we investigated the distribution of nELAVL, RO60 as well as Y RNAs upon UV exposure using cell fractionation followed by western blot and qPCR analysis. The induction of a UV stress response was confirmed by measuring *CDKN1A* mRNA levels (*Figure 7—figure supplement 3A*). We did not observe a change in nucleocytoplasmic localization of the investigated RNAs or proteins (*Figure 7—figure supplement 3B/C*), suggesting that the increased nELAVL/Y RNA association upon UV exposure does not result from a difference in the nucleocytoplasmic distribution of nELAVL or Y RNAs. These results are consistent with previous observations that neuronal ELAVL proteins show a higher cytoplasmic localization than the ubiquitous paralog ELAVL1 (*Kasashima et al., 1999*), and that stress-induced nuclear-cytoplasmic shuttling might be limited to ELAVL1 (*Burry and Smith, 2006*). Nonetheless, these results do not rule out the possibility that there may be changes of nELAVL proteins within the nuclear or cytoplasmic compartments themselves with respect to Y RNA binding and localization.

We next sought to measure the proportion of nELAVL bound to Y RNAs in stressed and non-stressed conditions. Because Y RNAs are relatively short and have a high degree of similarity, our mapping strategy (reporting only unambiguous mapping events) discarded numerous reads that were assigned to multiple Y RNAs. We therefore re-mapped CLIP tags, allowing multiple alignments, but reporting only the best match, permitting a more accurate estimate of overall Y RNA binding. The fraction of the short CLIP reads mapping to Y RNAs was considerably higher using this strategy, revealing that up to 6% of nELAVL CLIP tags map to Y RNAs in AD and UV stressed cells, compared to less than 0.5% in control brain and ~1% in non-stressed cells (*Figure 7—figure supplement 4*).

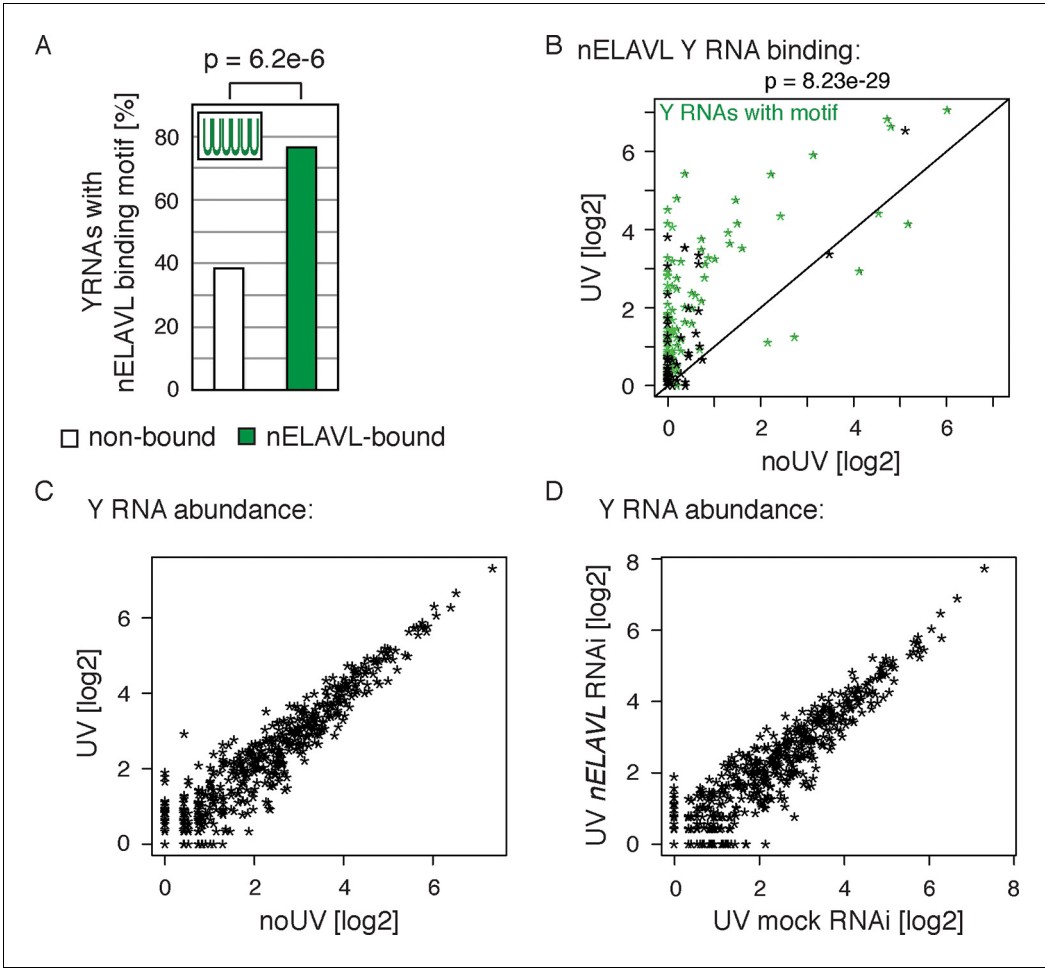

**Figure 7.** Y RNPs are remodeled during UV stress. (**A**) The nELAVL binding motif (UUUUUU, allowing a G at any position) is enriched in nELAVL-bound Y RNAs compared to non-bound Y RNAs (p = 6.2e⁻⁶; Fisher's exact test). Y RNAs were scanned for (T)$_6$, allowing a G at any position. nELAVL-bound Y RNAs: nELAVL CLIP tags in at least two samples; n = 132. (**B**) nELAVL binding of Y RNAs increases during UV stress compared to non-stressed cells (p = 8.23e⁻²⁹; paired one-sided Wilcoxon rank sum test). The axes depict nELAVL Y RNA binding (nELAVL CLIP tags per Y RNA) in control and UV stressed cells. Y RNAs with nELAVL binding motif are colored in green. (**C**) Y RNA levels do not change upon UV stress. Y RNA abundance (RNAseq tags per Y RNA) in UV stressed cells was plotted against Y RNA abundance in non-stressed control cells. (**D**) nELAVL is binding is not required for Y RNA stability. Comparison of Y RNA abundance between mock and *nELAVL* RNAi treated UV stressed cells.

The following figure supplements are available for figure 7:

**Figure supplement 1.** Y RNAs with a motif that are not bound are not expressed.

**Figure supplement 2.** UV-stressed cells show increased nELAVL binding to Y RNAs.

**Figure supplement 3.** UV does not induce changes in the nucleocytoplasmic localization of Y RNP components.

**Figure supplement 4.** Up to 5% of nELAVL CLIP map to Y RNAs in AD_Y subjects (AD subjects with increased nELAVL/Y RNA association) and UV stressed cells when mapped with Bowtie 2 (allowing multiple alignments and reporting one).

The significant increase in nELAVL/Y RNA association and our observation that up to 6% of nELAVL was bound to Y RNAs might in fact lead to a sequestration of nELAVL from its targets.

## nELAVL/Y RNA association correlates with loss of nELAVL-mediated splicing

To investigate if the increased nELAVL/Y RNA association was linked to decreased intronic and 3'UTR nELAVL binding, we grouped AD subjects based on their Y RNA association and compared the two different AD groups to control subjects. We found that the majority of changing nELAVL binding sites decreased in AD subjects with high nELAVL/Y RNA, while nELAVL binding in AD subjects with low nELAVL/Y RNA association was mostly increasing (*Figure 8A*). Because nELAVL binding in UV-stressed cells also predominantly decreased (*Figure 8A*) and most of the decreased binding sites were in introns (85%, assessed by annotation of peak locations), we speculate that nELAVL/Y RNA association leads to a sequestration of nELAVL specifically from its intron targets, which might induce similar splicing changes as *nELAVL* depletion by RNAi. Of note is our observation that nELAVL binding decreased at only a subset of intronic binding sites.

We further examined the possibility of Y RNA mediated nELAVL sequestration upon UV stress by subjecting mock and *nELAVL* RNAi treated IMR-32 cells to UV stress and analyzing these cells by RNAseq. *nELAVL* mRNA and proteins levels decreased upon *nELAVL* RNAi in non-stressed and UV stressed cells but were not affected by UV stress (*Figure 8—figure supplement 1* and *Supplementary file 2C*). We analyzed exon inclusion rates as above (*Figure 4*) and found that 9397 cassette exons were expressed between the four conditions (*Supplementary file 2E*). Comparing UV-induced splicing changes between mock and *nELAVL* RNAi treated cells, we identified 260 cassette exons that showed a differential inclusion rate upon UV stress only in the presence of nELAVL (*Figure 8B*). We intersected these splicing changes with *nELAVL* RNAi induced splicing changes (n = 553), and found a significant overlap between the two lists (n = 68; p = 9.7e$^{-28}$; hypergeometric test; *Figure 8C* and *Supplementary file 2E*). Importantly, splicing of the vast majority of exons (66 out of 68) changed in the same direction, indicating that changes in nELAVL binding due to UV stress partially recapitulate *nELAVL* RNAi depletion. This finding is consistent with a model of UV-induced nELAVL sequestration from a subset of targets.

Due to the wide difference between AD subjects and UV stressed IMR-32 cells, the targets affected by nELAVL sequestration in the two systems are likely to be markedly divergent. We nevertheless explored if any of the 66 affected exons in IMR-32 cells were adjacent (+/- 2.5 kb) to intronic nELAVL binding sites in human brain, and observed that 7 alternatively spliced exons were indeed next to intronic nELAVL binding sites (*Figure 8D/ Figure 8—figure supplement 2*). Remarkably, nELAVL binding at all of these sites decreased in AD subjects with Y RNA association but not in AD subjects without Y RNA association (*Figure 8D*), although nELAVL peak binding changed significantly at only one binding site (boxed in *Figure 8E*, FDR = 0.003). We also investigated if decreased nELAVL peak binding was associated with corresponding splicing changes in AD subjects (*Figure 8—figure supplement 2*). The inclusion rate of only one of the 7 exons changed significantly in AD patients with high nELAVL/Y RNA association (FDR = 0.04), and the direction of the splicing change was indeed the same as the splicing changes observed upon UV or *nELAVL* RNAi treatment. This is consistent with our observation that only few splicing changes are shared between AD and UV treatment, which likely reflects the much more complex situation in human brain.

## Y RNA overexpression is linked to nELAVL sequestration from mRNA targets

To directly test the model of Y RNA mediated nELAVL sequestration, we overexpressed canonical Y3wt (wild type) and Y3mut (with a mutated nELAVL binding site) using lentiviral infections of IMR-32 cells. We initially confirmed the overexpression of the infected Y RNAs using qPCR, which was more pronounced in the Y3mut infected cells (*Figure 9A*). The distribution of neither nELAVL nor RO60 was affected upon infection (*Figure 9—figure supplement 1*). To evaluate the extent of Y3wt and Y3mut overexpression compared to endogenous Y3 RNA expression we additionally analyzed infected cells by RNAseq (*Figure 9B*, *Figure 9—figure supplement 2*, *Supplementary file 3F*). While we consistently observed an increase in the Y3-like RNA expression upon infection, the magnitude of overexpression was modest relative to the endogenous expression of Y RNA copies. Nonetheless, we observed a small increase in total Y3-like RNAs in Y3wt but not Y3mut infected cells (*Figure 9B*).

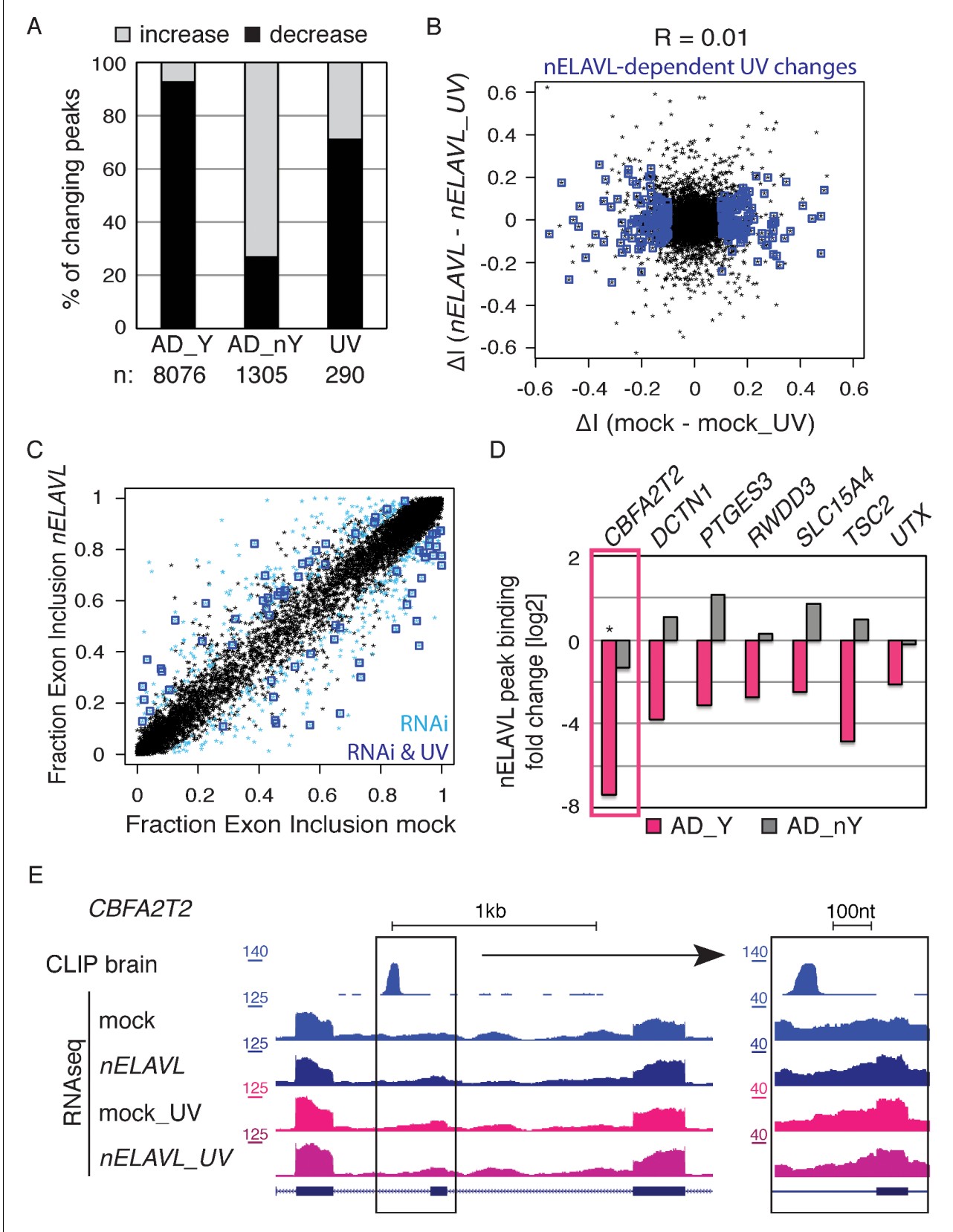

**Figure 8.** nELAVL/Y RNA correlates with loss of nELAVL-mediated splicing. (**A**) Samples with high nELAVL/Y RNA association show decreased nELAVL binding on mRNA targets. Columns represent significantly changing nELAVL binding sites. Shown are changes in AD subjects with and without Y RNA

*Figure 8 continued*

association (AD_Y and AD_nY) and changes upon UV treatment. The number of nELAVL binding sites (n) within each category is indicated. (**B**) Identification of nELAVL-dependent UV-induced splicing changes. Comparison of the differential inclusion rate of expressed cassette exons upon UV stress between mock and *nELAVL* RNAi treated IMR-32 cells (n = 9397). Significant UV-induced splicing changes that do not change upon UV stress in nELAVL RNA treated cells are boxed in dark blue (FDR<0.05 and ΔI>0.1; n = 260). (**C**) Many exons that are alternatively spliced upon nELAVL RNAi treatment also change during UV stress in an nELAVL-dependent manner. Shown is the inclusion rate of expressed cassette exons in IMR-32 cells that were subjected to mock or *nELAVL* RNAi (n = 9397). nELAVL RNAi induced splicing changes are colored in light blue (n = 553), and are additionally boxed in dark blue if they are UV-induced in an nELAVL-dependent manner (n = 68). The plot is related to *Figure 4A* but contains additional cassette exons expressed in UV stressed cells. (**D**) nELAVL binding adjacent to exons that are alternatively spliced upon *nELAVL* RNAi and UV treatment decreases only in AD subjects with an increased Y RNA association. Displayed is the change in nELAVL peak binding. nELAVL peak binding changes were not significant except for *CBFA2T2* (boxed in pink). * FDR<0.05 (derived from edgeR). (**E**) UCSC Genome Browser images depicting an overview and an enlarged view of a cassette exon within the *CBFA2T2* gene that is alternatively spliced in *nELAVL* RNAi and UV-treated IMR-32 cells. The nELAVL binding track in human brain and RNAseq tracks in mock and *nELAVL* RNAi treated non-stressed and UV-stressed IMR-32 cells are shown.

The following figure supplements are available for figure 8:

**Figure supplement 1.** UV does not affect nELAVL RNA or protein levels.

**Figure supplement 2.** Analysis of splicing changes in *nELAVL* RNAi and UV treated IMR-32 cells and AD subjects with and without Y RNA association (AD_Y and AD_nY).

We next investigated mRNA abundance and splicing changes upon Y RNA overexpression (*Supplementary file 2C/E*). The mRNA abundance changes upon Y3wt and Y3mut infection compared to non-infected controls were very similar, indicating that most mRNA abundance changes are due to lentiviral infection (*Figure 9—figure supplement 3A;* 70% of Y3wt changes overlapped with Y3mut changes; p = 1.3e$^{-175}$; hypergeometric test). To investigate virus-independent changes, we focused on the changes between Y3wt and Y3mut infection. nELAVL 3'UTR targets were not enriched among mRNAs that changed between Y3mut compared to Y3wt infected cells (*Figure 9—figure supplement 3B*), which is in agreement with our hypothesis that nELAVL sequestration predominantly affects intronic nELAVL binding sites and thus nELAVL mediated splicing.

In contrast to the mRNA abundance changes, only few splicing changes overlapped between Y3wt and Y3mut infection when compared to non-infected cells (17% of Y3wt induced changes overlapped with Y3mut induced changes). Most of the observed splicing changes are therefore likely to be specific to Y RNA overexpression. Importantly, we observed an enrichment of nELAVL bound exons and of *nELAVL* RNAi dependent exons among the exons that changed upon Y3wt but not Y3mut overexpression (*Figure 9C/D* and *Figure 9—figure supplement 4*, *5*). The relatively small enrichment is consistent with the modest increase in total Y3-like Y RNAs. These results suggest that Y RNA overexpression results in nELAVL sequestration from some of its intronic targets and consequent splicing changes, and partially recapitulates the stress induced nELAVL sequestration due to increased nELAVL/Y RNA association seen in AD patients and UV treated IMR-32 cells.

## Discussion

nELAVL proteins are abundant neuron-specific RNA binding proteins which have been suggested to regulate various neurological processes and have been linked to neurodegenerative disorders including AD and PD. Yet the RNA targets of nELAVL in human brain were completely unknown. Here, we generated a comprehensive genome-wide RNA binding map of nELAVL in human brain, identifying 75,592 significant binding events within 8681 transcripts. We observed a significant overlap between these binding sites and disease-associated 3'UTR SNPs, and the potential disruption of nELAVL-mediated RNA regulation at these sites might contribute to disease manifestation. Most deleterious variants to date have been identified by exome sequencing while as many as 50% of disease-causing mutations are thought to affect splicing (*Ward and Cooper, 2009*). With whole genome sequencing being increasingly available, non-coding variants are also increasingly detected, some of which may be linked to disease. As the majority of nELAVL binding occurs in introns and 3'UTRs, we expect that many binding sites will overlap with prospective disease-associated non-coding variants. The overlap between deleterious variants and nELAVL binding sites, and the observation that nELAVL binding at individual sites diverged between mice and human, underscores the

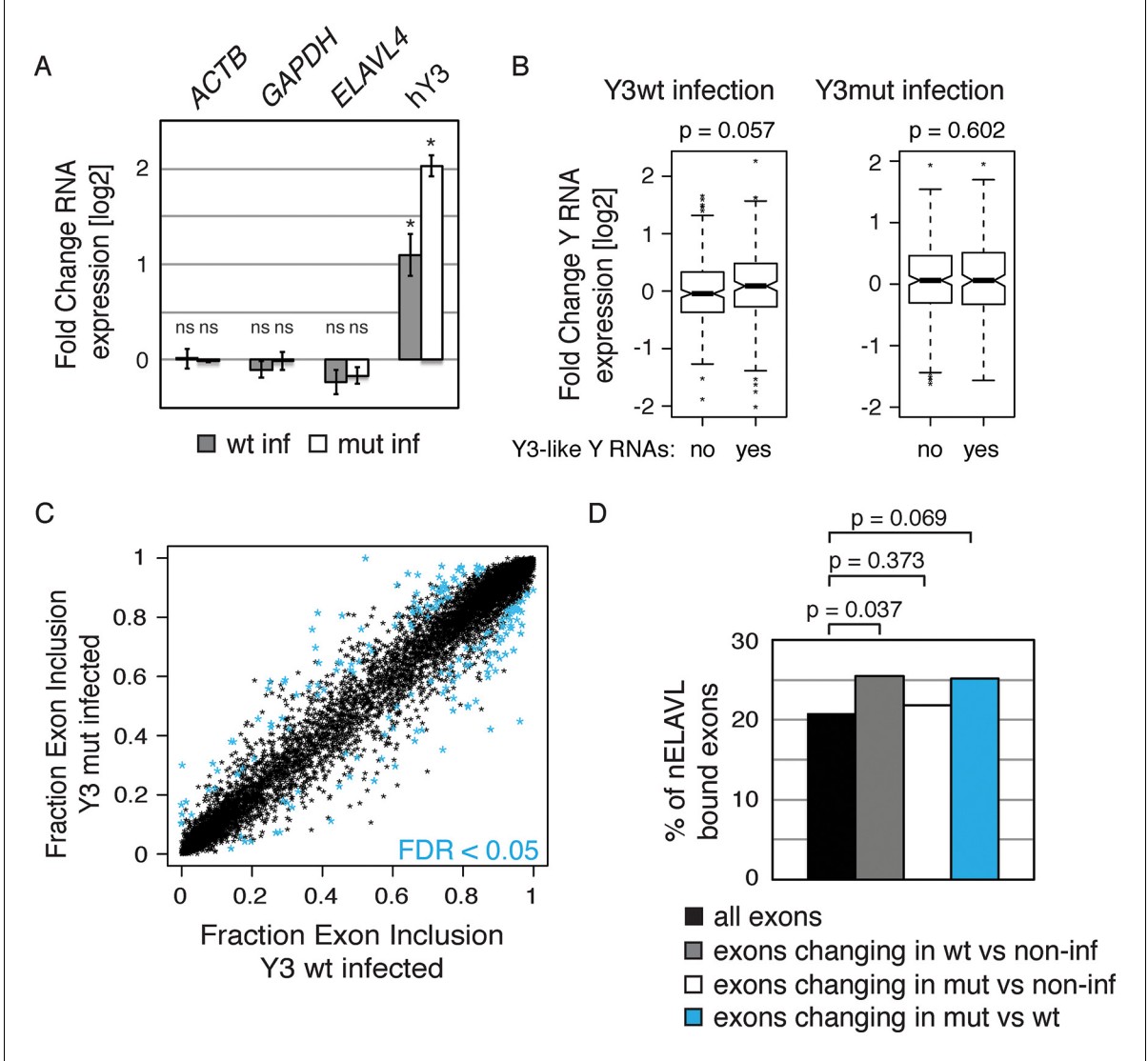

**Figure 9.** Y RNA overexpression is linked to nELAVL sequestration from mRNA targets. (**A**) Validation of Y RNA overexpression. Shown are RNA expression fold changes of Y3wt or Y3mut infected IMR-32 cells compared to non-infected IMR-32 cells assessed by qPCR. Y RNAs expression increased while control mRNAs (*ACTB, GAPDH, ELAVL4*) were not affected. Error bars represent SEM. p values were calculated with a two-tailed t-test (ns: not significant; * p<0.05). (**B**) The expression of endogenous Y3-like Y RNAs increases upon Y3wt but not Y3mut infection. Box plots represent the distribution of endogenous Y3-like and non-Y3-like Y RNA expression fold changes upon Y3wt or Y3mut infection. Y3-like Y RNAs show a slight increase in abundance upon Y3wt compared to non-Y3-like Y RNAs (p = 0.057; one-tailed t-test). In contrast, the mRNA abundance of Y3-like Y RNAs does not change upon Y3mut infection, when compared to non-Y3 like Y RNAs (p = 0.602; one-tailed t-test). (**C**) Identification of Y3 dependent splicing changes. Shown is the exon inclusion fraction of cassette exons that are expressed in IMR-32 cells subjected to Y3wt or Y3mut infection (n = 10,189). Exons changing significantly between Y3wt and Y3mut infection (FDR<0.05 and ΔI>0.1) are colored in light blue (n = 191). (**D**) Exons that are alternatively spliced upon Y3wt infection are enriched for nELAVL bound exons. Bar graph representing total expressed exons (n = 10,189), exons that change in either Y3wt (n = 240; blue points in the left panel of *Figure 9—figure supplement 4*) or Y3mut (n = 151; blue points in the right panel of *Figure 9—figure supplement 4*) infected cells compared to non-infected cells, and exons that change in Y3wt compared to Y3mut infected cells (n = 191; blue points in *Figure 9C*). Exons that are alternatively spliced upon Y3wt infection compared to either non-infected (p = 0.037; hypergeometric test) or Y3mut infected cells (p = 0.069; hypergeometric test) are enriched for nELAVL bound exons.

The following figure supplements are available for figure 9:

**Figure supplement 1.** Y3 overexpression does not induce changes in protein distribution.

**Figure supplement 2.** Validation of Y RNA overexpression.

*Figure 9 continued on next page*

*Figure 9 continued*

**Figure supplement 3.** Y3 overexpression does not lead to nELAVL 3'UTR target sequestration.

**Figure supplement 4.** Identification of Y3wt and Y3mut dependent splicing changes.

**Figure supplement 5.** Exons that are alternatively spliced upon Y3wt infection are enriched for *nELAVL* RNAi dependent exons.

importance of this study and illustrates the caveat of relying solely on mouse models when studying human disease. Considering the widespread nature of nELAVL binding in human brain and that RNA dysregulation has been linked to numerous neurological disorders, we believe that this binding map will be a valuable resource for the scientific community.

To analyze the functional consequences of nELAVL binding, we used two different loss-of-function models: *Elavl3/4* KO mice and nELAVL RNAi depletion in neuroblastoma cells. Due to the incomplete RNAi depletion of nELAVL in neuroblastoma cells, and potential differences in mRNA abundance and therefore nELAVL binding between the different samples, it is likely that we validated only a fraction of nELAVL-regulated transcripts. Despite these technical limitations we demonstrated that nELAVL impacts mRNA abundance and/or splicing of hundreds of targets. Among the nELAVL regulated transcripts were many transcripts implicated in human disease, including AD, which led us to investigate RNA regulation in AD subjects. Due to the relatively small sample size and the heterogeneity between these samples, likely due to both differences between individuals and sample preservation during postmortem collection, we did not detect many reproducible changes in mRNA abundance or nELAVL binding between AD and non-AD subjects. However, we found that 150 transcripts were differentially spliced in AD subjects, which in some cases coincided with differential nELAVL binding. Unexpectedly, the most significant binding change in AD was a dramatic increase in nELAVL binding to a class of non-coding RNAs, termed Y RNAs. This change was evident on a specific subset of Y RNAs harboring the nELAVL binding site. nELAVL/Y RNA binding also increased during UV stress in human neuroblastoma cells, while the abundance of Y RNAs remained constant in AD subjects and upon UV exposure. The increased nELAVL/Y RNA association correlated with decreased nELAVL binding at a subset of intronic binding sites, and was associated with similar splicing changes as induced by nELAVL depletion, suggesting that nELAVL/Y RNP remodeling during acute and chronic stress sequesters nELAVL from its mRNA targets. We provided further evidence for a Y RNA dependent nELAVL sequestration by overexpressing Y3 RNAs harboring either a wild type or mutated nELAVL binding site. Exons that were differentially spliced upon Y RNA overexpression were enriched for nELAVL bound exons, indicating nELAVL sequestration, which was dependent on an intact nELAVL binding site in the Y RNA.

nELAVL 3'UTR binding has been implicated in increasing mRNA abundance in vivo (*Ince-Dunn et al., 2012*). We described numerous nELAVL 3'UTR targets in brain, and were able to validate many of these targets, including disease-associated transcripts, indicating that nELAVL 3'UTR binding is important for the regulation of mRNA abundance in human brain. While ELAVL binding is frequently reported to result in an increase in mRNA abundance, we found several cases where nELAVL binding seemed to have an opposing effect. ELAVL proteins can compete or collaborate with miRNAs as well as RBPs like AUF1, CUGBP1 and TIA1 to regulate its targets (*Bhattacharyya et al., 2006*; *Kawai et al., 2006*; *Lal et al., 2004*; *Young et al., 2009*; *Yu et al., 2013*; *Kim et al., 2009*). The ultimate outcome of nELAVL 3'UTR binding might therefore vary between individual transcripts.

nELAVL has also been shown to regulate splicing in mouse brain by binding to intronic sequences (*Ince-Dunn et al., 2012*). We observed many instances of intronic nELAVL binding events adjacent to alternative exons in brain, and confirmed that nELAVL regulates many of these exons in mice and neuroblastoma cells. In contrast to the position-dependent splicing observed for other RBPs (*Licatalosi and Darnell, 2010*), we observed that upstream nELAVL binding was associated with both exon skipping and inclusion. While nELAVL binding was observed within 25-50 nucleotides upstream of skipped exons, coinciding with the branch point sequence, nELAVL binding peaked within the proximal 25 nucleotides upstream of included exons, overlapping the polypyrimidine tract. Binding of auxiliary splicing factors, including nELAVL, to the branch point sequence usually

interferes with spliceosome assembly and thus leads to exon skipping (*Licatalosi and Darnell, 2010*). Polypyrimidine tract binding however can lead to both exon inclusion and skipping (*Licatalosi et al., 2012*; *Wei et al., 2012*), presumably depending on the recruitment of splicing enhancers or silencers. Our data indicates that upstream nELAVL binding can both interfere with the assembly of the spliceosome as well as promote splicing, most likely by recruiting splicing enhancers.

Splicing defects have been associated with many neurological diseases (*Licatalosi and Darnell, 2006*), and among the nELAVL-regulated transcripts we describe here are numerous transcripts related to disease, including AD. For example, intronic nELAVL binding of the gene encoding the amyloid precursor protein, APP, was associated with skipping of exons 7 and 8. Both exons have previously been shown to be alternatively spliced and encode for the Kunitz protease inhibitory (KPI) motif, a domain that has been linked to APP processing (*Ben Khalifa et al., 2012*). Remarkably, KPI domain containing isoforms of APP have been shown to be increased in AD (*Zhang et al., 2012*), indicating that APP splicing might contribute to AD pathogenesis, and that nELAVL binding in human brain might be important to regulate the inclusion of the KPI domain. nELAVL regulates the splicing of two more AD-related transcripts, PICALM and BIN1, by promoting the inclusion of alternative exons 13 and 6a, respectively. Both proteins have been implicated in APP trafficking and both exons lie within domains mediating protein-protein interactions (*Tan et al., 2013*; *Treusch et al., 2011*). Moreover, inclusion of the alternative exon 13 in PICALM has been linked to an AD-associated SNP (*Parikh et al., 2014*), and we observed in this study that exon 6a of BIN1 shows a higher inclusion rate in controls compared to AD subjects. Since nELAVL binding promotes the inclusion of this exon, and control subjects show higher nELAVL binding, we propose that the altered splicing of BIN1 in AD subjects might be due to differential nELAVL binding. In fact, several nELAVL-regulated exons have been shown to be differentially spliced in AD subjects, further strengthening the link between nELAVL dysregulation and AD.

While Y RNAs have not been linked to AD before, they have been implicated in various types of stress responses. The RNA binding protein RO60 usually associates with Y RNAs and is required for their stabilization (*Chen et al., 2000*; *2003*; *Labbé et al., 1999*; *Wolin et al., 2013*; *Xue et al., 2003*). Besides RO60, Y RNPs contain several other RBPs such as ZBP1, MOV10, and Y-box proteins, and have been found to be remodeled upon stress (*Sim et al., 2012*). Our data suggests that nELAVL becomes increasingly associated with specific Y RNAs during both UV-induced stress and AD. ELAVL proteins can shuttle between nucleus and cytoplasm in response to environmental cues and preferentially accumulate in cytoplasmic stress granules upon cellular stress (*Fan and Steitz, 1998a*; *Gallouzi et al., 2000*), and ELAVL binding to the CAT-1 transcript is modulated in response to stress in cultured cells (*Bhattacharyya et al., 2006*). Interestingly, while we found that nELAVL specifically associates with Y RNAs during AD and acute UV stress, the nucleocytoplasmic distribution of nELAVL, RO60, and Y RNAs was not affected by UV stress. Because Y RNA levels remained constant, we propose that Y RNP complexes are specifically remodeled during AD and acute stress, which is not likely due to a change in nucleocytoplasmic protein/RNA distribution. These results are consistent with previous observations that stress induced shuttling might be limited to ELAVL1 (*Burry and Smith, 2006*). Our observation of Y RNP remodeling in two very different systems of neuronal stress suggests that differential nELAVL/Y RNA association may be a widespread phenomenon and a focus of future studies.

In addition to the four canonical human Y RNAs, hY1/3/4/5, hundreds of additional Y RNA genes are distributed throughout the human genome (*Perreault et al., 2005*). The apparent lack of promoters upstream led to a premature designation of these Y RNAs as pseudogenes. Surprisingly, we found that hundreds of these Y RNA copies are expressed in human brain and neuroblastoma cells, although it remains unclear if these Y RNAs can still associate with RO60, because the RO60 binding site in many Y RNA copies is mutated (*Perreault et al., 2005*). We observed that numerous Y RNA copies were more strongly associated with nELAVL in AD brain and acutely stressed cells, yet nELAVL binding did not affect their levels, indicating a function for this interaction other than Y RNA stabilization. While the outcome of nELAVL/Y RNA remains to be elucidated, our work revealed an aspect of nELAVL/Y RNA association related to stoichiometry. Hundreds of Y RNAs are bound by nELAVL in AD and UV-stress, which corresponds to up to 5% of all nELAVL CLIP tags. This shift of nELAVL binding may distort the normal stoichiometry of nELAVL interactions with its mRNA targets. Indeed, non-coding RNAs have previously been shown to affect RBP-RNA stoichiometry and

therefore the biological function of other RNAs or RBPs (*Borah et al., 2011*; *Cazalla et al., 2010*; *Hansen et al., 2013*). Our data indicate that the binding of nELAVL to Y RNAs during stress may lead to a redistribution of nELAVL binding and/or competition of nELAVL from other RNAs. Consistently, we found that high nELAVL/Y RNA association was associated with a general decrease in nELAVL binding at a subset of binding sites, especially within introns, and consequential splicing changes were reminiscent of splicing changes provoked by nELAVL depletion. Consistently, splicing changes induced by Y RNA overexpression showed an enrichment of nELAVL binding that was dependent on the presence of the ELAVL binding motif in Y RNAs. Hence we propose that the increased association of nELAVL and Y RNAs during stress causes sequestration of nELAVL from its mRNA targets.

Taken together, our data indicate that nELAVL becomes strongly associated with Y RNAs in some AD subjects as well as in cells subjected to UV stress, and this is linked to a sequestration of nELAVL from some of its intronic targets, partially recapitulating splicing changes induced by nELAVL depletion. Our results are consistent with a hypothesis that a relatively subtle and perhaps long-term effect of Y RNA binding on normal nELAVL stoichiometry may underlie subtle and long-term changes in nELAVL biology. Perhaps analogously, the sequestration of the RBP, TDP-43, has previously been linked to neurodegenerative disorders (*Lee et al., 2012*). While the underlying mechanisms of TDP-43 and nELAVL sequestration are distinct, relatively subtle and long-term rearrangement of RNA:protein stoichiometry and interactions might be a recurrent theme of neurodegeneration.

In conclusion, we have determined a robust and reproducible map of nELAVL binding sites on both mRNA and Y RNA targets in human brain. This database has both common and human-specific features that confirm and enhance previous work in mice and underscore its value as a resource for scientific inquiry. We have linked the data to genome-wide measurements of both mRNA levels and splicing suggesting specific functions for these important interactions. Moreover, we have uncovered a stress-modulated interaction of nELAVL proteins with non-coding Y RNAs. Y RNPs are remodeled during both UV-induced and chronic AD-related stress, which may be causally related to the pathophysiology of stress by causing a redistribution of nELAVL RNA target binding. Data implicating nELAVL proteins in several neurologic diseases, as well as the overlap of nELAVL binding sites we describe here with SNPs linked to the same and additional human diseases underscore the importance of this resource for the ongoing study of the molecular basis of human neurologic disease.

## Materials and methods

### Brain samples

Frozen brain tissue was obtained from the Mount Sinai Brain Bank. Subjects were classified according to CERAD criteria. Summaries of cognitive performance (assessed by clinical dementia rating, CDR), neurofibrillary tangles (assessed by Braak staging, BB) and plaque pathology (assessed by plaque counts) are shown in *Supplementary file 1A*. Brains were dissected along Brodmann areas and the two hemispheres were either stored as crushed powder at -80°C for biochemical analyses, or processed for stereological analyses, respectively. Eight control (no neurofibrillary tangles or plaque pathology), and nine advanced stage AD (CDR between 4 and 5) subjects matched for age and gender with short post mortem intervals (PMI) were selected. Crushed brain powder derived from the dorsolateral prefrontal cortex was either subjected to Trizol extraction or UV irradiation (3x 400 mJ/cm; see below) followed by CLIP analysis.

### Cell culture, nELAVL RNAi, and lentiviral Y3 overexpression

Human HEK293T and neuroblastoma IMR-32 cells were obtained from ATCC (Manassas, VA) and maintained in DMEM (Fisher Scientific, Pittsburgh, PA), containing 10% (v/v) fetal bovine serum (Thermo Scientific, Waltham, MA), and 100 unit/ml penicillin/streptomycin (Life Technologies, Carlsbad, CA). 0.25% trypsin and 1% EDTA (Invitrogen) were used to passage cells every three days. Prior to UV treatment, cells were transferred to poly-D-lysine-coated 10-cm culture dishes (BD Biosciences, San Jose, CA) and allowed to adhere for at least 24 hr. Cells were washed in PBS (Invitrogen), layered with 5 ml PBS, and exposed to UVC (254 nm; 0.2 and 0.5 mJ/cm$^2$) with 15 cm distance from the UV source using a Stratalinker. Cells were allowed to recover for 24 hr in fresh media

before being harvested. For RNAseq, cells were washed in 5 ml PBS, and frozen in 1 ml Trizol (Invitrogen) at -80°C. For CLIP, cells were washed in 5 ml PBS and irradiated at 400 mJ/cm$^2$ (see above). Cell pellets were collected by centrifugation at 2500 rpm for 3 min at 4°C and frozen at -80°C.

For *nELAVL* RNAi experiments, cells were to transferred to poly-D-lysine-coated 6-well plates (BD Biosciences, San Jose, CA) and allowed to adhere for at least 24 hr. Accell Non-targeting pool (Cat#D-001910-10-20), and a mixture of Accell siRNA pools against Elavl2/3 and 4 (Cat#E019801-00-0010, #E011264-00-0010, #E016006-00-0010) from Dharmacon (LaFayette, CO) were added to Accell siRNA Delivery Media (Cat#B-001910-10), supplemented with 2% FBS for a final concentration of 1 μM per pool. Growth medium was removed and 1 ml Accell siRNA and media mixture was added. The Accell siRNA and media mixture was replaced with growth medium after 48h, cells were UV treated at 72 hr (see above, cells were layered with 1 ml PBS during UVC exposure), and cells were harvested at 96 hr for RNAseq analysis. Cells were washed in 1 ml PBS and frozen in 1 ml Trizol (75%) or lysed (25%) in 50 μl CLIP Wash Buffer 1 (see below).

Canonical hY3 and hY3mut (sequences are shown below and mutated nucleotides are indicated) were cloned with *Age*I and *Eco*RI into the transfer plasmid plKO.1 (*Moffat et al., 2006*; Addgene [Cambridge, MA] #10878). The puromycin selection cassette was replaced with GFP to monitor infection efficiency. HEK293T cells were transfected with transfer and packaging plasmids (of a three-plasmid lentivirus packaging system) at 80% confluency, using X-tremeGENE9 DNA transfection agent (Roche, Indianapolis, IN). Lentivirus containing supernatant was harvested after 24 hr, concentrated, aliquoted and stored at -80°C. IMR-32 cells were transduced with virus in the presence of polybrene (Sigma, St. Louis, MO) with supernatant of one 10 cm plate per 6 well (transductions were performed in triplicates). The infection efficiency was assessed after 48 hr (80% of cells were GFP-positive), cells were expanded and harvested after 72 hr for RNAseq analysis (one 6-well per 1 ml Trizol) and 96 hr for cell fractionation experiments.

### Y3wt

GCTGGTCCGAGTGCAGTGGTGTTTACAACTAATTGATCACAACCAGTTACAGATTTCTTTGTTCCT-TCTCCACTCCCACTGCTTCACTTGACTAGCCTTTT

### Y3mut

GCTGGTCCGAGTGCAGTGGTGT<u>C</u>TACAACTAATTGATCACAACCAGTTACAGAT<u>C</u>TCT<u>CC</u>GTTCC-TTCTCCACTCCCACTGCTTCACTTGACTAGCCTTTT

## Cell fractionation, qPCR and Western blot analysis

Cells grown in 10-cm culture dishes were scraped in 1 ml PBS after media removal and spun at 500 *g* for 5 min at 4°C. Cells were lysed in 500 μl buffer A (150 ml NaCl; 50 mM Tris, pH 7.4; 0.01% Saponin; 1x Protease Inhibitor Cocktail [Thermo Fisher Scientific, Waltham, MA]), and incubated on ice for 10 min. Cytoplasm and nuclei were separated with a 10 min spin at 3000 *g* at 4°C, and both fractions were washed in 500 μl buffer A before an additional 10 min spin at 3000 *g* at 4°C. Nuclei were resuspended in 500 μl buffer A and sonicated 3x for 5 s. RNA was Trizol (Life Technologies) extracted from 100 μl lysate and reverse transcribed using iScript (Bio-Rad, Hercules, CA). qPCR was performed with FastStart SYBR Green Master (Roche, Indianapolis, IN), requiring at least one primer in each mRNA primer pair to be specific for an exon junction. qPCR results were normalized as indicated. Western blot analysis was performed using 15 μl cell lysate per lane, and the following antibodies: α-GAPDH (1:25:000; Abcam [Cambridge, MA] ab8245), Hu subject antiserum (1:1000 dilution), α-HSP90 (1:1000; Cell Signaling Technology [Danvers, MA] 4877S), α-H3 (1: 2000; Abcam ab1791), α-RNA PolII (1:1000; Millipore [Billerica, MA] 05–623), and α-RO60 (1:50; gift from S. Wolin). Immunoreactive bands were analyzed with the Odyssey Infrared Imaging System (LI-COR, Lincoln, NE) and normalized as indicated.

## nELAVL HITS CLIP

nELAVL HITS CLIP was performed as previously described (*Ule et al., 2005*) with the following modifications. nELAVL-RNP complexes were immunoprecipitated using paraneoplastic Hu-antiserum, which recognizes all three neuronal ELAVL isoforms. 80 mg human brain powder or one 10-cm 70% confluent plate of IMR-32 cells were used per immunoprecipitation. For controls (no UV irradiation,

control serum, and overdigested control), 40 mg of human brain powder or half a mouse brain were used, respectively. Prior to phosphatase treatment, beads were washed for 3 min each in a series of wash buffers (see below). nELAVL bound RNA fragments from two IMR-32 samples were ligated to an indexed degenerate 5′ RNA linker (see *Supplementary file 1A*); RNA fragments from the remaining samples were ligated to a degenerate 5′ RNA linker. The two-step PCR amplification was performed with Accuprime Pfx (Invitrogen). cDNA from IMR-32 and subject samples was initially amplified with DP3/5. Brain cDNA from subjects 1–3, and 9–11 was then amplified with DSFP3/5, samples 4–8, and 12–17 were amplified with MSFP3 and indexed MSFP5. Brain samples were sequenced on an Illumina (San Diego, CA) GAIIX at the Rockefeller University Genomics Resource Center with the standard Illumina primer (samples 4–8, and 12–17) or the custom primer SSP1 (samples 1–3, and 9–11). IMR-32 cDNA was either amplified with SP3/5-PE (samples with indexed 5′linker), and sequenced on the MiSeq system with the custom primer SSP1, or with MSFP3 and indexed MSFP5 and sequenced on the MiSeq system with the standard Illumina primer. See *Supplementary file 1A* for used indexes.

## Wash buffers
Wash buffer 1: 1xPBS; 0.1%SDS; 0.5%Na-DOC; 0.5% NP-40
  Wash buffer 2: 5xPBS; 0.1%SDS; 0.5%NA-DOC; 0.5% NP-40
  Wash buffer 3: 15 mM Tris, pH7.4; 5 mM EDTA; 2.5 mM EGTA; 1% Triton X-100; 1% Na-DOC; 0.1% SDS; 120 mM NaCl; 25 mM KCl
  Wash buffer 4: 15 mM Tris, pH7.4; 5 mM EDTA; 2.5 mM EGTA; 1% Triton X-100; 1% Na-DOC; 0.1% SDS; 1 mM NaCl
  Wash buffer 5: 15 mM Tris, pH7.4; 5 mM EDTA
  Wash buffer 6: 50 mM Tris, pH7.4; 10 mM MgCl$_2$; 0.5% NP-40

## Linker and primer sequences
3′ RNA linker: 5′P – GUG UCA GUC ACU UCC AGC GG 3′ – puromycin
  5′ RNA linker: 5′OH – AGG GAG GAC GAU GCG GNN NNG 3′ – OH
  5′ RNA linker indexed: 5′OH – AGG GAG GAC GAU GCG GXX NNN NG 3′ – OH
  DP5: 5′ – AGG GAG GAC GAT GCG G – 3′
  DP3: 5′ – CCG CTG GAA GTG ACT GAC AC – 3′
  DSFP5: AATGATACGGCGACCACCGACTATGGATACTTAGTCAGGGAGGACGATGCGG
  DSFP3: CAAGCAGAAGACGGCATACGACCGCTGGAAGTGACTGACAC
  SP5-PE:                          AATGATACGGCGACCACCGAGATCTACACCTATGGATACTTAGTCAGGGAGGACGATGCGG
  SP3-PE: CAAGCAGAAGACGGCATACGAGATCTCGGCATTCCTGCCGCTGGAAGTGACTGACAC
  MSFP3: CAAGCAGAAGACGGCATACGAGATCCGCTGGAAGTGACTGACAC
  MSFP5: AATGATACGGCGACCACCGAGATCTACACTCTTTCCCTACACGACGCTCTTCCGATCT-XXXXAGGGAGGACGATGCGG
  SSP1: CTATGGATACTTAGTCAGGGAGGACGATGCGG

## Analysis of CLIP samples
Analyses were carried out using the Galaxy suite of bioinformatics tools (http://main.g2.bx.psu.edu/), in addition to publicly available in-house tools. Data was visualized with UCSC genome browser (http://genome.ucsc.edu/). To reduce mis-alignments due to sequencing errors, reads were initially filtered based on quality score ($\geq$20 in the degenerate linker region; average of $\geq$20 in the remaining read). Exact sequences were collapsed to remove PCR duplicates. The degenerate barcode (and index if applicable) were removed and the 3′ linker was trimmed. Using FASTA files as input, reads were subsequently mapped to the hg18 build of the human genome by novoalign (www.novocraft.com), requiring unambiguous mapping and a maximum of two mismatches. To identify unique CLIP tags, we applied stringent filtering, and collapsed reads with the same starting genomic positions (*Darnell et al., 2011*), and only unique tags (*Supplementary file 1A*) were used for subsequent analyses. Peaks with significant nELAVL binding compared to background (p-value<0.01) were identified utilizing a similar approach from previous studies (*Xue et al., 2009*). Specifically, we used a

scan statistics (*Glaz et al., 2013*) to compute p-values in which each observed PeakHeight (PH, CLIP tags within a peak) was compared to the PH one would expect by random chance, assuming a background of uniformly distributed CLIP tags in each gene. Bonferroni correction was applied to the resulting p-values to correct for multiple hypothesis testing. All scripts used in the analysis including the peak finding algorithm and more information can be publically obtained at http://zhanglab. c2b2.columbia.edu/index.php/Resources.

## RNAseq library preparation and analysis

RNA from human brain and IMR-32 neuroblastoma cell lines was Trizol (Invitrogen)-extracted, Ribo-Zero-selected (Epicentre, Madison, WI), DNase-treated (Roche), and prepared for sequencing, following the Illumina High-throughput TruSeq RNA Sample Preparation Guidelines. The libraries from subjects 6–8 and 15–17, as well Y RNA overexpression samples, were sequenced on an Illumina HiSeq 2500 at the New York Genome Center, generating 125-bp paired end reads. The remaining libraries were sequenced on an Illumina HiSeq 2000 at the Rockefeller University Genomics Resource Center, yielding 100-bp paired end reads. Reads were aligned to the hg18 build of the human genome using TopHat, allowing a maximum of two mismatches. Only unambiguously mapped reads were kept for analysis. Additional exon junctions not observed due to the gap between paired end mates were inferred using a Bayesian method, and a set of non-redundant constitutive exons with relative high inclusion rate (according to ESTs) was used to estimate gene expression (*Charizanis et al., 2012*). Exon and exon junction reads were inferred as previously described (*Charizanis et al., 2012*). 8163 cassette exons in brain, 9629 cassette exons in the RNAi/UV IMR-32 samples, and 10,432 cassette exons in the Y RNA infected IMR-32 samples remained after filtering on exon junction coverage to reduce multiple testing (coverage was normalized for library size; normalized coverage of each isoform and each condition $\geq 5$).

## Downstream CLIP and RNAseq analysis and statistics

CLIP and RNAseq data were deposited in the GEO database with accession number GSE53699. Analyses of unique CLIP tags and RNAseq data were carried out in R (www.r-project.org; [*Ihaka and Gentleman, 1996*]) using Bioconductor (*Gentleman et al., 2004*) and the packages Biostrings, edgeR, GenomicRanges, ggplot2, Hmisc, plyr, qvalue, reshape, scales, and VennDiagram (*Robinson and Smyth, 2007*; *2008*; *Robinson et al., 2010*; *McCarthy et al., 2012*; *Harrell et al., 2014*; *Wickham, 2007*; *2009*; *2011*; *2014*; *Aboyoun et al., 2014*; *Pages et al., 2014*; *Dabney, et al., 2014*; *Chen and Boutros, 2011*). mRNA abundance and nELAVL peak binding (PeakHeight, PH) changes were assessed by differential analysis of raw sequencing counts in edgeR using the TMM methodology (*Robinson and Oshlack, 2010*) for normalization and a negative binomial generalized linear model. Only expressed genes and robustly bound peaks were analyzed to reduce multiple testing. Expressed genes were defined as genes that had more than one count per million (cpm) in at least 5 control brain samples (out of 8), in 6 AD brain samples (out of 9), or in 4 IMR-32 samples (out of 12 RNAi/UV samples or 9 Y RNA infection samples, respectively). Robustly bound peaks were defined as peaks that had more than one cpm in at least 5 control or AD brain samples (out of 8 and 9, respectively), or in 2 control or 4 UV IMR-32 samples (out of 3 and 6, respectively). We controlled for batch effects of brain samples (see *Supplementary file 1A*) and estimated the false discovery rate (FDR) with an optimized FDR approach (q-value methodology [*Storey and Tibshirani, 2003*]) to correct for multiple hypothesis testing. PeakHeight (PH, CLIP tags within peaks) was normalized for library size, and, to account for differences in gene expression level, normalized PH was determined by dividing PH by rpkm of the corresponding gene. nELAVL binding was defined as the summary of PH per gene, whereas normalized nELAVL binding was defined as the summary of normalized PH per gene. We similarly defined 3'UTR and intronic binding by summarizing only 3'UTR or intronic peaks, respectively. Top targets were defined as the top 1000 genes according to normalized nELAVL binding. mRNA abundance was defined as cpm RNAseq tags per gene. We did not normalize nELAVL binding nor mRNA abundance for transcript length as we performed differential analysis of the datasets. A pseudocount of 1 was added before log2 transformation. Cross-correlation plots show log2 raw counts of PH or RNAseq reads per gene. Splicing changes were determined using the observed inclusion and exclusion junction read counts and by fitting a generalized linear model with a logit link function. For each exon, GLM likelihood ratio test was conducted

to test if there was a significant difference in the fraction of exon inclusion (delta Inclusion, ΔI) between conditions, and controlling for batch variables of brain samples. ΔI (equivalent to delta PSI) was calculated by subtracting fraction exon inclusion of Sample 2 from fraction exon inclusion of Sample 1. FDR was calculated as described before using the q-value method. High confidence alternative splicing events were identified by requiring a stringent criteria of FDR<0.05 and ΔI≥0.1. P-values to assess the statistical significance of the overlap between gene lists were calculated with a hypergeometric test (*Figure 2A/9D*, *Figure 9—figure supplement 5*). A paired Wilcoxon rank sum test was used to evaluate differences in Y RNA binding (*Figure 6C/7B*), and a t-test was used to evaluate changes in mRNA/Y RNA/protein abundance (*Figure 3B/9A,B*, *Figure 3—figure supplement 1*, *Figure 6—figure supplement 1*, *Figure 7—figure supplement 1*, *3*,*Figure 8—figure supplement 1A*, *Figure 9—figure supplement 1*, *2*). Differences in cumulative density curves were evaluated with a one-sided KS (Kolmogorov-Smirnov) test (*Figure 3B*). P-values to assess motif enrichment were calculated with a Fisher's exact test (*Figure 6B/7A*).

## Y RNA analysis

Y RNAs were derived from GENCODE (v19 release). nELAVL binding on Y RNAs was defined as CLIP tags on Y RNAs. To discriminate between individual Y RNA copies we used CLIP reads, aligned with novoalign. While this mapping strategy allowed us to discrimate between individual Y RNA copies with high confidence, CLIP reads that mapped to multiple Y RNAs were discarded. Exclusively for *Figure 7—figure supplement 4*, CLIP tags were therefore additionally aligned with Bowtie2 (*Langmead et al., 2009*), allowing multiple alignments and reporting one. This allowed us to estimate the overall amount of CLIP tags on all Y RNAs (shown in *Figure 7—figure supplement 4*). Y RNA abundance in brain, and UV- and RNAi-treated IMR-32 cells was defined as RNAseq tags on Y RNAs, using unambiguously mapped tags aligned with TopHat. Similar to above, reads that mapped to multiple Y RNAs were discarded using this strategy. To estimate overall Y RNA overexpression upon lentiviral infection, we mapped RNAseq reads of infected samples with Bowtie2, allowing multiple alignments and reporting the best matched alignment (*Figure 9B*). Y RNA overexpression was determined with two additional strategies. We performed qPCR with primers that recognized infected Y RNAs (Y3wt and Y3mut) but also an unknown number of endogenous Y RNAs (*Figure 9A*). Additionally, we directly searched for 40- and 68-nt sequences of Y3wt and Y3mut sequences in the Illumina sequenced library reads (both 40- and 68-nt sequences spanned the mutated residues). The 68-nt sequence was only present in the infected Y RNAs and the canonical hY3 RNA, and searching for this sequencing length showed that both Y3wt as well as Y3mut were overexpressed. To more accurately estimate the extent of overexpression, we additionally searched for a 40-nt Y RNA sequence. The Y3wt sequence is present in a large number of endogenous Y RNAs, whereas the Y3mut sequence only detects infected Y3mut. By comparing Y3wt and Y3mut reads in Y3mut infected samples, we observed that infected Y RNA reads (Y3mut reads) correspond to 10% of total Y3 RNA reads (Y3wt reads).

## Motif analysis

The MEME-ChIP Suite (*Machanick and Bailey, 2011*) and HOMER (*Heinz et al., 2010*) were used for motif discovery. Coordinates of the top 500 peaks in healthy brain +/- 25 nt were used as input. The given strand was searched for 6–10-nt-long motifs for up to 10 motifs, allowing any number of repetitions with MEME-ChIP. We additionally searched the given strand with HOMER v4.7 using default parameters and 50 nt flanking sequences of the 500 peaks as background sequences. The respectively top enriched motif is shown.

## Protein-protein interaction network analysis

The 1000 nELAVL top target gene products were seeded in a large protein-protein interaction (PPI) network containing 14,191 genes and more than 197,000 interactions (*Neale et al., 2012*) compiled from various online databases: BioGrid (*Stark et al., 2011*), MINT (*Ceol et al., 2010*), KEGG (*Aoki and Kanehisa, 2005*), PPID (*Hermjakob et al., 2004*), HPRD (*Prasad et al., 2009*), DIP (*Xenarios et al., 2002*), BIND (*Isserlin et al., 2011*), IntAct (*Aranda et al., 2010*), InnateDB (*Lynn et al., 2008*), and SNAVI (*Ma'ayan et al., 2009*). To reduce the false positive interactions in the background network, we excluded interactions reported using high-throughput approaches.

Direct interactions with other seed genes were kept for each seed gene, and a connected subnetwork was created using these seed genes as nodes. Organic clustering of the obtained subnetwork was performed using the network visualization software yEd (http://www.yworks.com/en/products_yed_about.html). In a second network, six AD genes (APP, BACE1, MAPT, PICALM, PSEN1 and PSEN2) were concatenated with the input seed gene lists to investigate their relationship with nELAVL target genes. The composition and the clustering of the PPI networks with and without those six genes were identical. Only the network including those additional seeds is shown.

### Enrichment analysis

The clusters of nELAVL target genes in the PPI subnetwork represent protein clusters based on direct physical interactions. Gene products in clusters with at least 10 nodes were examined for an enrichment of functional annotations with Enrichr (*Chen et al., 2013*) using the Fisher's exact test and computing an adjusted p-value for multiple hypothesis testing using the Benjamini-Hochberg correction. The six additional AD associated genes used for the subnetwork clustering were not taken into account for the enrichment analysis.

## Acknowledgements

We are indebted to members of the Darnell lab for comments on the manuscript, Samuel Gandy, Dimos Gaidatzis, Lukas Burger, Michael Stadler, Jernej Ule, and Nicolas Robine for helpful discussions, Shih-Chen Fu for support related to PPI network analysis, and Shengdong Ke, and Chaolin Zhang for their bioinformatic expertise.

## Additional information

### Author contributions

CS, performed experiments and bioinformatic analyses, interpreted the data and wrote the paper ; ED, performed experiments and wrote the paper ; MAF, JF, AM, performed experiments; CYP, provided bioinformatics support; IZS, performed experiments ; YK, performed PPI network analysis ; VH, provided brain tissue; JDB, designed the study, interpreted the data, and wrote the paper ; RBD, designed the study, interpreted the data, and wrote the paper

### Funding

| Funder | Grant reference number | Author |
| --- | --- | --- |
| EMBO | Long-Term Fellowship | Claudia Scheckel |
| SNF | Early and Advanced Postdoc Mobility Fellowship | Claudia Scheckel |
| Seaver Foundation | Graduate Fellowship | Yan Kou |
| National Institutes of Health | ADRC AG005138 | Vahram Haroutunian |
| National Institutes of Health | AG02219 | Vahram Haroutunian |
| National Institutes of Health | HHSN27120130031C | Vahram Haroutunian |
| National Institutes of Health | R01GM098316 | Avi Ma'ayan |
| National Institutes of Health | U54CA189201 | Avi Ma'ayan |
| National Institutes of Health | U54HL127624 | Avi Ma'ayan |
| G Harold and Leila Y. Mathers Foundation | | Joseph D Buxbaum |
| National Institutes of Health | ADRC AG005138 | Joseph D Buxbaum |
| National Institutes of Health | NS081706 | Robert B Darnell |
| Simons Foundation | | Robert B Darnell |
| National Institutes of Health | NS34389 | Robert B Darnell |

Howard Hughes Medical                                     Robert B Darnell
Institute

The funders had no role in study design, data collection and interpretation, or the decision to
submit the work for publication.

# Additional files

## Supplementary files

• Supplementary file 1. Supplementary files 1A-F. (A) Sample summary. Shown is subject sample information including postmortem interval (PMI), age, pathology diagnosis, cognitive performance assessed by clinical dementia rating (CDR), plaque pathology assessed by plaque counts, and neuritic performance assessed by Braak staging (BB), as well as read statistics, used linker indexes, and batch information for RNAseq and CLIP data for brain and IMR-32 samples. (B) RNAseq data from human brain List containing 14,876 genes that are expressed in either control or AD brain. Shown are Entrez Gene ID and Symbol, rpkm, log cpm, and the comparison between control and AD brain using edgeR (log2 average cpm, log2 FC (fold change), p and FDR values). (C) nELAVL binding sites in human brain. The table contains 74,423 peaks in human brain that mapped to transcripts expressed in brain. Shown are PeakName, hg18 and hg19 genomic coordinates (chromosome, start, end, strand), Entrez Gene ID and Symbol, genomic region, BC (biological complexity), PH (PeakHeight), and PH normalized for mRNA abundance. (D) nELAVL targets in human brain. Shown are 8681 genes with nELAVL binding sites in human brain. Entrez Gene ID and Symbol, rpkm, log2 cpm and the peak number, log2 nELAVL binding (CLIP tags within peaks were summarized per transcript), and log2 normalized nELAVL binding (CLIP tags within peaks were normalized for mRNA abundance and summarized per transcript) of total, 3'UTR and intronic peaks are displayed. Top targets and top 3'UTR targets can be extracted by selecting the top 1000 targets according to log2 normalized nELAVL binding or log2 normalized 3'UTR nELAVL binding, respectively. (E) nELAVL top targets in human brain. List of top 1000 nELAVL targets in human brain. Entrez Gene ID and Symbol, rpkm, log2 cpm and the peak number, log2 nELAVL binding (CLIP tags within peaks were summarized per transcript), and log2 normalized nELAVL binding (CLIP tags within peaks were normalized for mRNA abundance and summarized per transcript) of total, 3'UTR and intronic peaks are displayed. Information on the validation of nELAVL-mediated regulation is included if applicable. (F) PPI Cluster analysis of top nELAVL targets in human brain. Enriched terms of nELAVL top target gene clusters in the PPI subnetwork. The Benjamini-Hochberg corrected p value is < 0.05 for all terms. For large gene clusters (≥20 genes), enriched terms containing no less than 4 genes from the cluster are shown in the table; for small gene clusters (<20 genes), enriched terms containing no less than 2 genes from the cluster are shown. Terms are ranked by increasing p value and only the top 10 terms with a p value <0.05 are shown.

• Supplementary file 2. Supplementary files 2A-F. (A) nELAVL 3'UTR targets validated in mice. Displayed are 37 genes that change significantly in their mRNA abundance in *Elavl3/4* KO mice and that have 3'UTR binding in human brain. Included are mouse Entrez Gene ID and Symbol, the count of mouse 3'UTR CLIP tags, mRNA level log2 FC (fold change) in *Elavl3/4* KO mice (KO/ WT), and t-test results. Human Entrez Gene ID and Symbol, log2 cpm and the peak number, log2 nELAVL binding (CLIP tags within peaks were summarized per transcript), and log2 normalized nELAVL binding (CLIP tags within peaks were normalized for mRNA abundance and summarized per transcript) of 3'UTR peaks are displayed. (B) nELAVL intron targets validated in mice. Table containing exons that are alternatively spliced in *Elavl3/4* KO mice and that have intronic binding adjacent to them (+/- 2.5kb) in human brain. Shown are mouse Gene Symbol, mouse exon information (mm9 and mm10 coordinates), the difference in the exon inclusion rate in *Elavl3/4* KO mice compared to wilt-type (∆I: WT-KO), adjacent mouse CLIP tags, information on experimental validation, human Entrez Gene ID and Symbol, human exon information (hg18 and hg19 coordinates), human peak information (name, hg18 and hg19 coordinates, PH (PeakHeight), PH normalized for mRNA abundance, and the position of the peak relative to the alternative exon (negative values indicate that the

peak is upstream of the exon). (C) RNAseq data from IMR-32 cells. List containing 13100 genes that are expressed in at least one IMR-32 dataset (RNAi, UV stress, Y3 infection). Shown are Entrez Gene ID and Symbol, rpkm and log cpm for the different conditions, and the comparisons between the different conditions using edgeR (log2 average cpm, log2 FC (fold change), p and FDR values). If applicable, nELAVL binding information (peak number, log2 nELAVL binding (CLIP tags within peaks were summarized per transcript), and log2 normalized nELAVL binding (CLIP tags within peaks were normalized for mRNA abundance and summarized per transcript) in human brain) is included. (D) nELAVL 3'UTR targets validated in IMR-32 cells. List containing 743 genes expressed in brain that are changing in their abundance upon *nELAVL* RNAi and their nELAVL 3'UTR binding if applicable. Included are Entrez Gene ID and Symbol, rpkm and log2 cpm in mock and *nELAVL* RNAi treated cells, log2 average cpm, log2 FC (fold change), p and FDR values comparing the mRNA abundance between mock and *nELAVL* RNAi treated cells. Additionally, rpkm and log2 cpm in human brain, and the peak number, log2 nELAVL binding (CLIP tags within peaks were summarized per transcript), and log2 normalized nELAVL binding (CLIP tags within peaks were normalized for mRNA abundance and summarized per transcript) of 3'UTR peaks (if applicable) in human brain are shown. Top 3'UTR targets are indicated in in the last column. Transcripts that decrease upon *nELAVL* RNAi can be obtained by selecting rows with log2FC_MockElavl <0. (E) Splicing data from IMR-32 cells. List containing 13496 alternative exons that are expressed in at least one IMR-32 dataset (RNAi, UV stress, Y3 infection). Shown are AS event information (Name, hg18 and hg19 genomic coordinates of the AS event and the alternative exon), Entrez Gene ID and Symbol, exon inclusion fraction, and the comparisons between different conditions (the change in exon inclusion (ΔI), p and FDR values). Peak information (Name, hg18 and hg19 genomic coordinates, PH (PeakHeight), PH normalized for mRNA abundance, and the peak position relative to the alternative exon peaks) of peaks adjacent (+/- 2.5kb) to alternative exons is shown. Information on exons that change in different conditions can be found in the last 7 columns. (F) nELAVL intron targets validated in IMR-32 cells. Displayed are 473 alternative splicing (AS) events that change upon *nELAVL* RNAi, and if applicable adjacent intronic nELAVL binding sites. Shown are AS event information (Name, hg18 and hg19 genomic coordinates of the AS event and the alternative exon), Entrez Gene ID and Symbol, alternative exon inclusion in mock and *nELAVL* RNAi (PSI), the change in exon inclusion (ΔI), p and FDR values comparing PSI between mock and *nELAVL* RNAi treated cells, and peak information if applicable. This includes Peak Name, hg18 and hg19 genomic coordinates, PH (PeakHeight), PH normalized for mRNA abundance, and the position of the peak relative to the alternative exon (negative values indicate that the peak is upstream of the exon).

• Supplementary file 3. Supplementary files 3A-F. (A) Overlap of nELAVL 3'UTR targets and genes with disease associated 3'UTR SNPs. List containing transcripts with nELAVL 3'UTR binding with disease associated 3'UTR SNPs (245 SNPs mapped to 200 genes). Included are Entrez Gene ID and Symbol, rpkm and log2 cpm, nELAVL 3'UTR binding information (peak number, log2 nELAVL binding (CLIP tags within peaks were summarized per transcript), and log2 normalized nELAVL binding (CLIP tags within peaks were normalized for mRNA abundance and summarized per transcript) in human brain), SNP information (coordinates, associated disease), and, if applicable, information on overlapping peaks (name, hg18 and hg19 coordinates, PH (PeakHeight), PH normalized for mRNA abundance). (B) CLIP and RNAseq data of AD-related genes List of 96 AD-related genes. Shown are Entrez Gene ID and Symbol, rpkm and log2 cpm, and the peak number, log2 nELAVL binding (CLIP tags within peaks were summarized per transcript), and log2 normalized nELAVL binding (CLIP tags within peaks were normalized for mRNA abundance and summarized per transcript) of peaks (if applicable) in human brain. Top targets are indicated in in the last column. (C) Overlap of top nELAVL targets and GWAS-associated AD genes. Displayed are 77 top nELAVL targets with AD-associated SNPs (p< 0.001). Entrez Gene ID and Symbol, rpkm and log2 cpm, and the peak number, log2 nELAVL binding (CLIP tags within peaks were summarized per transcript), and log2 normalized nELAVL binding (CLIP tags within peaks were normalized for mRNA abundance and summarized per transcript) of peaks (if applicable) in human brain are shown. (D) nELAVL binding sites in control and AD subject brain. The table contains 113,671 nELAVL binding sites that are bound in control or AD brain and that present within expressed genes. Included are PeakName, hg18 and hg19 genomic coordinates (chromosome, start, end, strand), Entrez Gene ID and Symbol, genomic region, PH

(PeakHeight), PH normalized for mRNA abundance, and BC (biological complexity) in control and AD brain, and log2 cpm (counts per million CLIP reads per peak), log2 FC (fold change), p and FDR values comparing PH between control and AD brain. (E) Splicing changes in AD Displayed are 170 alternative splicing (AS) events that change in AD. Shown are AS event information (Name, hg18 and hg19 genomic coordinates of the AS event and the alternative exon), Entrez Gene ID and Symbol, exon inclusion fraction in control and AD brain (PSI), the change in exon inclusion (ΔI), p and FDR values comparing PSI between control and AD brain. Among the differentially spliced exons were five nELAVL regulated exons (validated in IMR-32 cells), which were adjacent to 8 intronic peaks). For those exons, alternative exon inclusion in mock and *nELAVL* RNAi (PSI), the change in exon inclusion (ΔI), p and FDR values comparing PSI between mock and *nELAVL* RNAi treated cells, and information of adjacent peaks (Name, hg18 and hg19 genomic coordinates, biological complexity (BC), PH (PeakHeight), PH normalized for mRNA abundance, and the peak position relative to the alternative exon) is additionally included. (F) Y RNA expression and nELAVL binding in brain and IMR-32 cells. Table containing 864 Y RNAs. Shown are Y RNA information (Name, hg18 and hg19 Y RNA coordinates, the presence of 6mer or 5mer Hu binding motif, and type), and log2 average of YRNA sequencing (seq) and nELAVL binding (CLIP) in brain and IMR-32 cells, and the YRNA expression log2 fold change upon Y3wt and Y3mut infection. Y RNAs bound in AD and IMR-32 cells can be extracted by selecting BC_brain_CLIP$\geq$2 and BC_IMR-32_CLIP $\geq$ 2, respectively.

### Major datasets

The following datasets were generated:

| Author(s) | Year | Dataset title | Dataset URL | Database, license, and accessibility information |
|---|---|---|---|---|
| Scheckel et al | 2015 | nELAVL mediated RNA regulation during Alzheimer's disease and UV stress | http://www.ncbi.nlm.nih.gov/geo/query/acc.cgi?acc=GSE53699 | Publicly available at the NCBI Gene Expression Omnibus (Accession no: GSE53699) |

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
