## [Decision Letter]

Thank you for submitting your work entitled "Regulatory consequences of nELAVL binding to coding and non-coding RNAs in human brain" for peer review at *eLife*. Your submission has been favorably evaluated by a Senior Editor, and two reviewers, one of whom, Douglas Black, is a member of our Board of Reviewing Editors.

The reviewers have discussed the reviews with one another and the Reviewing editor has drafted this decision to help you prepare a revised submission.

This paper from the Darnell group examines the RNA binding sites of the nELAVL family in human brain. The authors perform CLIP analysis and RNAseq in the BA9 region of the prefrontal cortex. They identify a large set of RNA binding sites and presumed target transcripts. They define the nELAVL binding motif as similar to the previously defined motif for the mouse and show that there is strong overlap between the targets seen in mouse and human. They show that nELAVL knockdown in human neuroblastoma cells causes reduced expression of mRNAs that exhibit 3' UTR binding. Similarly, exons with adjacent nELAVL binding show altered splicing with knockdown. Interestingly, both positively and negatively regulated exons exhibit enriched binding upstream, and thus do not present the same correlation of binding position with positive or negative regulation as seen with some other splicing regulators. Earlier results had made a connection of nELAVL proteins and Alzheimer's disease, and the authors next analyzed splicing and nELAVL binding in AD brains, finding some exons and binding events that differ between AD and control brains. The most significant change was a greatly increased level of nELAVL binding to a subset of Y RNAs in the AD brains. Y RNAs are a class of small noncoding RNAs in eukaryotic cells that affect RNA scavenging and quality control pathways. A subset of these RNAs contain an nELAVL binding site. Cellular stress is known to relocalize ELAV proteins to the cytoplasm, and they find that stressing tissue culture cells with low UV doses also increases nELAV/Y RNA binding as measured by CLIP. This is accompanied by changes in nELAV dependent splicing, presumably due to reduced nuclear protein.

This is a novel study of a topic with broad interest. The nELAV proteins have been widely studied. The extension of these analyses to the human system and to AD is significant, with the connection of nELAVL proteins to the Y RNAs being potentially very important. The authors present a large amount of work and the datasets generated will be a valuable resource for further studies. However, the analysis does not extend past identifying broad correlations between datasets, leaving the biological or mechanistic conclusions unclear. A number of unaddressed issues regarding the identified nELAV/Y RNA interaction make its significance hard to judge. The paper is also difficult to digest. It is hard to follow what was actually done in the various statistical comparisons, what the results mean, and the authors conclusions often seem vague. They often point to rather weak correlations as being "consistent" with a particular model. Sometimes they use abbreviations and acronyms that don't seem to have been defined (for example, is dI the same as delta PSI?). There are a number of places where the analytical methods used could be improved or at least better explained.

Major issue:

1) The most significant finding in the study is the binding of nELAVL to the Y RNAs, but this is not taken far enough to draw many conclusions. Is it the increased Y RNA binding that is causing the change in splicing or is it a consequence of increased cytoplasmic protein that is the result of cellular stress? How much Y RNA is there relative to nELAVL? What percent of the nELAVL is actually bound by the Y RNAs? Are the amounts sufficient to sequester most of the ELAVL and thus have an effect? Why do so many Y RNAs without ELAVL binding motifs also appear to be binding? Can the authors connect the splicing or expression changes observed in the UV treated IMR32 cells to those seen in AD? Without this, it is difficult to make the case that the cytoplasmic relocalization of the nELAVL proteins is relevant to AD. The reviewers all felt that the authors should work to make some of these mechanistic connections between the Y RNAs and AD and thus strengthen this most interesting finding.

Issues regarding the analytical methods:

1) The peak finding in the CLIP analysis needs to be better described. The authors refer to Licatalosi (2012), but that paper describes a relatively outmoded method of defining peaks above a gene specific background, as well as the use of the much better CIMS analysis from the Darnell and Zhang groups. It doesn't appear that CIMS analysis was used here, so what exactly was done?

2) Similarly, MEME-ChIP has been shown to not work well for RBP motif finding. Stating that the motif identified in human is in excellent agreement with that found in mouse is not very meaningful since both studies used the same MEME-ChIP analysis. The authors should investigate the use of newer tools such as Graphprot (Maticzka et al. Genome Biology 2014) or Zagros (Bahrami-Samani et al., NAR, 2015) for motif discovery.

3) In Figure 1, are the peak numbers in different regions normalized for the lengths of the different types of sequences?

4) There are also normalization questions regarding Figure 1 and other plots. These plots show correlation between CLIP peaks and tags with mRNA abundance as measured by sequence reads per gene. This is problematic on several levels. First, both CLIP and RNAseq reads will be strongly affected by gene length. A better comparison is to use CLIP clusters per unit length of gene vs RPKM from RNAseq, which will normalize both values by gene length. Second even with length normalization, CLIP clusters will automatically vary strongly with gene expression level. A true comparison would need to look at the distributions of sequence density for genes in the CLIP and RNAseq datasets.

5) The authors use the human genome release hg18 for their alignments, although the conclusions do not likely depend on the genome release. Most people in the field are using the now 6-year-old hg19 and it would make it much easier for others to use these results if the authors were to update their pipelines to hg19.

---

## [Author Response]

*This paper from the Darnell group examines the RNA binding sites of the nELAVL family in human brain. The authors perform CLIP analysis and RNAseq in the BA9 region of the prefrontal cortex. They identify a large set of RNA binding sites and presumed target transcripts. They define the nELAVL binding motif as similar to the previously defined motif for the mouse and show that there is strong overlap between the targets seen in mouse and human. They show that nELAVL knockdown in human neuroblastoma cells causes reduced expression of mRNAs that exhibit 3' UTR binding. Similarly, exons with adjacent nELAVL binding show altered splicing with knockdown. Interestingly, both positively and negatively regulated exons exhibit enriched binding upstream, and thus do not present the same correlation of binding position with positive or negative regulation as seen with some other splicing regulators. Earlier results had made a connection of nELAVL proteins and Alzheimer's disease, and the authors next analyzed splicing and nELAVL binding in AD brains, finding some exons and binding events that differ between AD and control brains. The most significant change was a greatly increased level of nELAVL binding to a subset of Y RNAs in the AD brains. Y RNAs are a class of small noncoding RNAs in eukaryotic cells that affect RNA scavenging and quality control pathways. A subset of these RNAs contain an nELAVL binding site. Cellular stress is known to relocalize ELAV proteins to the cytoplasm, and they find that stressing tissue culture cells with low UV doses also increases nELAV/Y RNA binding as measured by CLIP. This is accompanied by changes in nELAV dependent splicing, presumably due to reduced nuclear protein. This is a novel study of a topic with broad interest. The nELAV proteins have been widely studied. The extension of these analyses to the human system and to AD is significant, with the connection of nELAVL proteins to the Y RNAs being potentially very important. The authors present a large amount of work and the datasets generated will be a valuable resource for further studies. However, the analysis does not extend past identifying broad correlations between datasets, leaving the biological or mechanistic conclusions unclear. A number of unaddressed issues regarding the identified nELAV/Y RNA interaction make its significance hard to judge. The paper is also difficult to digest. It is hard to follow what was actually done in the various statistical comparisons, what the results mean, and the authors conclusions often seem vague. They often point to rather weak correlations as being "consistent" with a particular model. Sometimes they use abbreviations and acronyms that don't seem to have been defined (For example, is dI the same as delta PSI?). There are a number of places where the analytical methods used could be improved or at least better explained.*

We have streamlined our analysis, added more details and explanation on comparisons and abbreviations in the manuscript, and included more information on the analysis in the Methods. We hope these changes will make the manuscript easier to follow.

Major issue:

*1) The most significant finding in the study is the binding of nELAVL to the Y RNAs, but this is not taken far enough to draw many conclusions. Is it the increased Y RNA binding that is causing the change in splicing or is it a consequence of increased cytoplasmic protein that is the result of cellular stress?*

The reviewers suggest two possibilities to account for the change in splicing upon stress. We addressed the possibility that a decrease in nuclear nELAVL might lead to the splicing changes in AD and upon UV stress. To address this point, we have added cell fractionation experiments. Figure 7—figure supplement 3 shows that the cytoplasmic fraction of nELAVL did not increase upon UV stress in IMR32 cells (using induction of CDKN1A as a positive control for the induction of UV stress). Similarly, the nucleocytoplasmic distribution of YRNAs did not change under the UV stress conditions we used (same Figure). This suggests that changes observed upon UV stress in IMR32 cells are not due to changes in the nucleocytoplasmic distribution of nELAVL. This does not address whether there may be changes of nELAVL accessibility within either of these compartments. We note that Burry and Smith (2006; Figure 1) did see relocalization of ELAVL1 (HuR) under conditions of stress, but did not see relocalization of ELAVL4 (HuD). Hence the biology of the different nELAVL isoforms may depend on many variables, including cell type and stress conditions. We discuss these points in the manuscript in the subsection “Y RNPs are remodeled during UV stress”.

To address the reviewers’ second possibility, we have included new data to examine the role of Y RNA induced splicing changes (see below).

*How much Y RNA is there relative to nELAVL? What percent of the nELAVL is actually bound by the Y RNAs? Are the amounts sufficient to sequester most of the ELAVL and thus have an effect?*

To address this point, we have estimated how much of the total amount of nELAVL is bound by Y RNAs under control versus stress conditions. Each Y RNA family (Y1/Y3/Y4/Y5) consists of multiple variants expressed from different genomic loci. These copies are relatively short and very similar, yet too different to be detected by one Taqman probe or one qPCR primer set. This makes absolute quantification of Y RNA levels difficult, and thus prevented us from quantifying the exact ratio of Y RNAs to nELAVL. Nonetheless, we were able to estimate that up to 6% of nELAVL was bound by Y RNAs, which is shown in Figure 7—figure supplement 4. To do so, we mapped our CLIP datasets with Bowtie2, allowing multiple mapping events and reporting the best one. The amount of nELAVL bound to Y RNAs was estimated by the percentage of total nELAVL CLIP tags on Y RNAs. We conclude that Y RNAs sequester nELAVL from only a subset of nELAVL targets (85% of binding sites decreasing in AD were intronic, as assessed by annotation of peak locations; please see the subsection “nELAVL/Y RNA association correlates with loss of nELAVL-mediated splicing”). Consistent with this observation, nELAVL-dependent mRNA abundance regulation was not affected, and the splicing of only a subset of nELAVL regulated exons correlated with nELAVL Y RNA binding. Taken together, our data are consistent with a hypothesis that a relatively subtle and perhaps long-term effect of YRNA binding on normal nELAVL stoichiometry may underlie subtle and long term changes in nELAVL biology. We clarified these points in the current version of the manuscript.

*Why do so many Y RNAs without ELAVL binding motifs also appear to be binding?*

We used a stringent definition of the nELAVL binding motif when searching Y RNA sequences (a stretch of 6 U’s, allowing a G at any position). Y RNAs that are bound by nELAVL and don’t have a motif, have a more degenerate version of the motif.

A total of 320 Y RNAs are bound by nELAVL, and we found the stringent definition of the nELAVL binding motif in 202 of these Y RNAs. We examined the 118 Y RNAs that did not fit this consensus in more detail. 91 (77%) of these Y RNAs contained the motif with an A or C instead of a G, or a 5mer version of the motif. The remaining 27 Y RNAs harbored UG rich stretches. We have added these points to the current version of the manuscript (subsection “Non-coding Y RNAs are bound by nELAVL in AD”).

*Can the authors connect the splicing or expression changes observed in the UV treated IMR32 cells to those seen in AD? Without this, it is difficult to make the case that the cytoplasmic relocalization of the nELAVL proteins is relevant to AD.*

The splicing changes in UV treated IMR-32 cells do not significantly overlap with the splicing changes observed in AD (subsection “nELAVL/Y RNA association correlates with loss of nELAVL-mediated splicing”). This is certainly in part due to the wide difference between the two systems that were analyzed. As noted above, the splicing changes do not seem to relate to changes in the cytoplasmic fraction of nELAVL proteins. We recognize that increased nELAVL/Y RNA association may lead to nELAVL sequestration within cellular compartments in human brain or IMR-32 cells, but we were not able to generate high quality data to address this issue. We recognize the limitations and potential interest of these observations for future studies and have added this to our manuscript (subsection “Y RNPs are remodeled during UV stress”).

*The reviewers all felt that the authors should work to make some of these mechanistic connections between the Y RNAs and AD and thus strengthen this most interesting finding.*

To prove that the increased nELAVL/Y RNA association in AD indeed sequesters nELAVL, we would have to manipulate Y RNA levels in AD patients. As this is impossible, we sought to manipulate Y RNA levels in IMR-32 cells. Since the expressed Y RNA copies are too diverse to be collectively targeted by, for example, shRNAs, we instead overexpressed Y RNAs to strengthen the mechanistic connection between Y RNAs and nELAVL sequestration. We infected IMR-32 cells with lentivirus containing Y3wt RNA (contains an nELAVL binding site) and Y3mut RNA (with a mutated nELAVL binding site), and analyzed changes in RNA regulation upon Y RNA infection. Because of similarities between different Y RNAs, we quantified the overexpression in several different ways, shown in Figure 9 and Figure 9—figure supplement 2 (qPCR, Bowtie2 mapping, and direct search of Y3wt and mut sequences in raw Illumina sequencing reads). Because of the high number of endogenous Y RNAs, exogenous Y RNA overexpression led to a relatively small change in total Y3-like RNAs in Y3wt but not Y3mut infected cells. This did not change nELAVL and RO60 protein distribution (shown in Figure 9—figure supplement 1). The small increase in the pool of Y3-like RNAs did induce small effects on splicing, while Y3mut infection did not.

Importantly, the differentially spliced exons upon Y3wt expression were enriched for exons with adjacent intronic nELAVL binding sites, and this enrichment was dependent on the presence of the nELAVL-binding site in the Y RNA (Figure 9). These results strengthen the mechanistic connection between Y RNA and nELAVL sequestration, and have been added to the subsection “Y RNA overexpression is linked to nELAVL sequestration from mRNA targets”.

We believe that the clarifications added to the manuscript and the addition of cell fractionation experiments, as well as an entire dataset describing the consequences of Y RNA overexpression, address the reviewers’ concerns and strengthen our conclusions.

*Issues regarding the analytical methods:*

*1) The peak finding in the CLIP analysis needs to be better described. The authors refer to Licatalosi (2012), but that paper describes a relatively outmoded method of defining peaks above a gene specific background, as well as the use of the much better CIMS analysis from the Darnell and Zhang groups. It doesn't appear that CIMS analysis was used here, so what exactly was done?*

While the analysis of CIMS (crosslinked-induced mutations) is indeed a further development of the initial peak analysis, CIMS analysis does not replace peak analysis. While CIMS analysis provides nucleotide resolution of RNA protein interactions, CIMS are only found in a small minority of peaks. To provide a more complete picture of nELAVL binding sites in the brain, we investigated nELAVL binding via peak analysis. We have added more information on the peak analysis in the Methods section.

*2) Similarly, MEME-ChIP has been shown to not work well for RBP motif finding. Stating that the motif identified in human is in excellent agreement with that found in mouse is not very meaningful since both studies used the same MEME-ChIP analysis. The authors should investigate the use of newer tools such as Graphprot (Maticzka et al. Genome Biology 2014) or Zagros (Bahrami-Samani et al., NAR, 2015) for motif discovery.*

We agree with the reviewers that a motif agreement solely based on MEME-ChIP would not be very meaningful. However, the motif identified in mice was verified via an independent method, in vitro binding assays (Ince-Dunn et al., 2012). We additionally followed the reviewers’ suggestion and compared a number of different tools, including the ones the reviewers suggested. We found the best independent tool to be HOMER (http://homer.salk.edu/homer/motif/), and used this to validate the motif identified with MEME-ChIP. The newly identified motif was very similar to the previously identified motif and is shown in Figure 1—figure supplement 2.

*3) In Figure 1, are the peak numbers in different regions normalized for the lengths of the different types of sequences?*

No, the peak numbers in Figure 1 are not normalized for region length. We have added Figure 1—figure supplement 3, where we show the peak distribution normalized for region length. As expected, the peak distribution changes significantly when normalized for regions length, and 3’UTRs show a far higher per-nucleotide density compared to other regions. The result has been added to the text.

*4) There are also normalization questions regarding Figure 1 and other plots. These plots show correlation between CLIP peaks and tags with mRNA abundance as measured by sequence reads per gene. This is problematic on several levels. First, both CLIP and RNAseq reads will be strongly affected by gene length. A better comparison is to use CLIP clusters per unit length of gene vs RPKM from RNAseq, which will normalize both values by gene length. Second even with length normalization, CLIP clusters will automatically vary strongly with gene expression level. A true comparison would need to look at the distributions of sequence density for genes in the CLIP and RNAseq datasets.*

We thank the reviewers for suggesting this way of normalization. We have now normalized each binding site for sequence density (rpkm), before summarizing binding sites per transcript (or per 3’UTR or introns). We have therefore redefined top targets (about two third of the top targets overlapped with previously defined top targets), as well as top 3’UTR in the manuscript.

*5) The authors use the human genome release hg18 for their alignments, although the conclusions do not likely depend on the genome release. Most people in the field are using the now 6-year-old hg19 and it would make it much easier for others to use these results if the authors were to update their pipelines to hg19.*

We agree with the reviewers that the use of the hg19 build would be more appropriate at this point. To address this point and make our datasets more useful to a wider audience, we added hg19 coordinates of all supplied genomic coordinates using the UCSC liftover tool. We hope the reviewers will agree with us that the amount of time it would take us to remap the large number of datasets, and re-do the entire analysis as well as all of the tables and figures, is not justified by the relatively minor improvement in data usage compared with the liftover we included in this revised version. We would also like to emphasize that all raw sequences have been deposited at GEO for future studies.